# TEM8 marks neovasculogenic tumor-initiating cells in triple-negative breast cancer

Jiahui Xu[1], Xiaoli Yang[1], Qiaodan Deng[1], Cong Yang[2], Dong Wang[3], Guojuan Jiang[1], Xiaohong Yao[4], Xueyan He[1], Jiajun Ding[1], Jiankun Qiang[1], Juchuanli Tu[1], Rui zhang[1], Qun-Ying Lei [1], Zhi-min Shao[1], Xiuwu Bian [4✉], Ronggui Hu [5✉], Lixing Zhang [1✉] & Suling Liu [1✉]

Enhanced neovasculogenesis, especially vasculogenic mimicry (VM), contributes to the development of triple-negative breast cancer (TNBC). Breast tumor-initiating cells (BTICs) are involved in forming VM; however, the specific VM-forming BTIC population and the regulatory mechanisms remain undefined. We find that tumor endothelial marker 8 (TEM8) is abundantly expressed in TNBC and serves as a marker for VM-forming BTICs. Mechanistically, TEM8 increases active RhoC level and induces ROCK1-mediated phosphorylation of SMAD5, in a cascade essential for promoting stemness and VM capacity of breast cancer cells. ASB10, an estrogen receptor ERα trans-activated E3 ligase, ubiquitylates TEM8 for degradation, and its deficiency in TNBC resulted in a high homeostatic level of TEM8. In this work, we identify TEM8 as a functional marker for VM-forming BTICs in TNBC, providing a target for the development of effective therapies against TNBC targeting both BTIC self-renewal and neovasculogenesis simultaneously.

[1] Fudan University Shanghai Cancer Center & Institutes of Biomedical Sciences; Cancer Institutes; Key Laboratory of Breast Cancer in Shanghai; The Shanghai Key Laboratory of Medical Epigenetics; The International Co-laboratory of Medical Epigenetics and Metabolism, Ministry of Science and Technology; Shanghai Medical College, Fudan University, Shanghai, China. [2] School of Medicine, Guizhou University, Guiyang, Guizhou, China. [3] WPI Nano Life Science Institute, Kanazawa University, Kakuma-machi, Kanazawa, Japan. [4] Institute of Pathology and Southwest Cancer Center, Southwest Hospital, Third Military Medical University (Army Medical University); Key Laboratory of Tumor Immunopathology, Ministry of Education of China, Chongqing, China. [5] State Key Laboratory of Molecular Biology; CAS Center for Excellence in Molecular Cell Science; Shanghai Institute of Biochemistry and Cell Biology, Chinese Academy of Sciences, Shanghai, China. ✉email: bianxiuwu@263.net; coryhu00@gmail.com; zhang_lx@fudan.edu.cn; suling@fudan.edu.cn

Breast cancer (BC) is the most common cancer in women. Approximately 10–20% of BC cases is negative for estrogen receptor (ER), progesterone receptor, and HER2 hormone receptors. This type of BC is defined as triple-negative breast cancer (TNBC), which is non-responsive to commonly used hormonal therapy medicines or medicines that target the HER2 protein[1,2]. A limited number of options are available to effectively treat TNBC.

The rapid growth of malignant solid tumors requires an efficient blood vessel network to provide enough oxygen and nutrition. Currently, clinical anti-angiogenesis approaches are predominantly anti-VEGF/VEGFR pathway therapies[3]. However, preclinical trials indicated that anti-angiogenesis therapies were only moderately effective and did not significantly prolong progression-free survival or overall survival in the primary setting, in addition to the cytotoxic side-effects associated with their pleiotropic activities[3–5]. This is in part due to the fact that the tumor vascular system is more complicated than expected. In addition to the VEGF pathway, alternate neovasculogenic pathways play vital roles in tumor progression by promoting the formation of a perfusable vasculogenic-like network called vasculogenic mimicry (VM), which is typically resistant to conventional anti-angiogenesis drugs[6]. BC cells with VM capacity contributed to primary tumor growth and drove distant metastasis[7]. Residual tumor cells after standard chemotherapies often readily formed VM channels in TNBC, leading to tumor relapse after treatment[8]. The existence of tumor-initiating cells (TICs) is currently proposed to underlie not only treatment failure, but also tumor recurrence and metastasis[9]. The plasticity of TICs also contributes to VM[10,11]. Breast tumor-initiating cells (BTICs) are highly heterogeneous[12,13]. ALDH+ BTICs had a greater capacity for VM than ALDH- cells in vitro[14]. Despite the critical importance of VM-forming TICs for the development, progression, and relapse of BC, the specific VM-forming BTIC population and the underlying regulatory mechanisms remain undefined.

Tumor endothelial marker 8 (TEM8) is a highly conserved integrin-like glycoprotein that was originally identified as a tumor endothelial marker based on its predominant expression in the vasculature of human malignant neoplastic tissues[15,16]. Among TEM family members, TEM8 was distinct in that it was preferentially expressed in tumor vessels and not detected in normal blood vessels. Knockout of the Tem8 gene did not affect the normal development and physiologic angiogenesis of mice, whereas it dramatically impaired the pathological angiogenesis and xenograft tumor growth in multiple cancer types[17,18]. This highlights the potential of TEM8 as an anti-tumor angiogenesis target. Although the loss of TEM8 in breast tumor cells inhibited tumor growth and metastasis in a mouse xenograft model[19,20], little is known about the functions of TEM8 in breast tumor-associated neovasculogenesis, especially in VM. Here, we report the molecular basis of the TEM8-mediated regulation of VM-forming BTICs and its contribution to neovasculogenesis and tumorigenesis in TNBC. In addition, we propose a regulatory mechanism underlying the control of TEM8 proteostasis.

## Results

**TEM8 promoted vasculogenic mimicry of TNBC cells**. Neovasculogenesis, which plays a critical role in TNBC progression, has not been studied extensively in BC. Here, we identified genes involved in the regulation of neovasculogenesis in BC by analyzing the genetic profiles of seven tumor tissue samples from TNBC patients and the corresponding adjacent non-cancerous (para-tumor) tissues, which were overlapped with two angiogenic gene clusters. Two genes were upregulated in common (Fig. 1a,

Supplementary Fig. 1a, b). We focused on the gene TEM8, which was predominantly expressed in the tumor endothelium[15].

Because the efficacy of commercial TEM8 antibodies is limited, we developed the polyclonal antibody RB9075 against the extracellular domain of TEM8 to investigate its involvement in BC. This antibody was suitable for use in western blotting (Supplementary Fig. 2a–d), flow cytometry (Supplementary Fig. 2e), immunohistochemistry (IHC, Supplementary Fig. 2g), and immunofluorescence (Supplementary Fig. 2h). RB9075 failed to recognize human CMG2, a protein with high similarity to TEM8 (Supplementary Fig. 2d, f), confirming its specificity for TEM8. IHC staining for TEM8 was performed in a cohort of 81 BC patient tumor tissues and 18 para-tumor tissues. The results showed that TEM8 was upregulated in BC cells, whereas few positive cells were detected in para-tumor tissues (Fig. 1b). TEM8 expression was higher in TNBC than in luminal and HER2+ subtypes (Supplementary Fig. 1c). We further investigated the relationship between TEM8 levels and clinical characteristics. TEM8 expression was inversely associated with ER status ($P < 0.001$), PR status ($P < 0.001$), and tumor histological grade ($P < 0.05$) (Supplementary Table 1). Consistently, analysis of BC cell lines showed high expression of TEM8 in TNBC (Supplementary Fig. 2a). TNBC patients with higher TEM8 expression had worse relapse-free survival (Supplementary Fig. 1d). Taken together, these results indicated that TEM8 was upregulated in BC cells, especially in TNBC, and might play a vital role in regulating BC malignant progression.

TEM8 plays a critical role in tumor-associated endothelial cells and tumor progression[18]. However, little is known about its function in neovasculogenic in BC cells especially TNBC cells. To assess the role of TEM8, we first analyzed the correlations between TEM8 and breast tumor-associated neovessel density. CD31 and Periodic Acid-Schiff (PAS) staining were used to differentiate endothelial microvessel (CD31+) from tumor cell-derived VM (CD31-PAS+)[7]. The results showed that tumor vessel density, especially VM density (VMD), was lowest in luminal BC and highest in TNBC (Fig. 1c–e); there was no significant correlation between tumor size and VMD (Supplementary Fig. 1e). TEM8hi BCs had higher microvessel density (MVD) and VMD than TEM8low BCs in both BC patient tumors and TNBC patient-derived xenograft (PDX) tumors (Fig. 1f, g). Staining of blood vessels with isolectin B4 allowed monitoring the transition between CD31+ endothelial vessels and CD31-TEM8+ non-endothelial vessels in TNBC patient tumors (Fig. 1h). Vessels generated from TNBC tumor cells expressed high levels of TEM8 (Fig. 1b, black arrowheads). MDA-MB-231lung, a MDA-MB-231 derived cell line with enhanced lung metastasis capability, also showed higher expression of TEM8 (Supplementary Fig. 3a). Consistent with the results obtained in BC samples, TEM8hi MDA-MB-231lung cells, sorted by fluorescent activated cell sorting (FACS) and confirmed by qRT-PCR (Supplementary Fig. 3b, c), showed stronger VM capacity in vitro (Supplementary Fig. 1f) and enhanced tumor growth ability in vivo (Supplementary Fig. 3d, e), suggesting a positive correlation between TEM8 and breast tumor neovasculogenesis, especially tumor cell VM capacity.

To verify the function of TEM8 in regulating tumor cell VM, stable TEM8-overexpressing and TEM8-knockdown TNBC cell lines were established (Supplementary Fig. 2b, c). TEM8 overexpression promoted breast tumor growth and metastasis (Supplementary Fig. 3f–h), whereas TEM8 knockdown had the opposite effects (Supplementary Fig. 3i–k), as previously reported[20]. Consistently, the results of in vitro tube formation assays showed that capillary-like structures were significantly increased in TEM8-overexpressing cells (Fig. 1i). Significant positive correlations were observed between TEM8 and VM

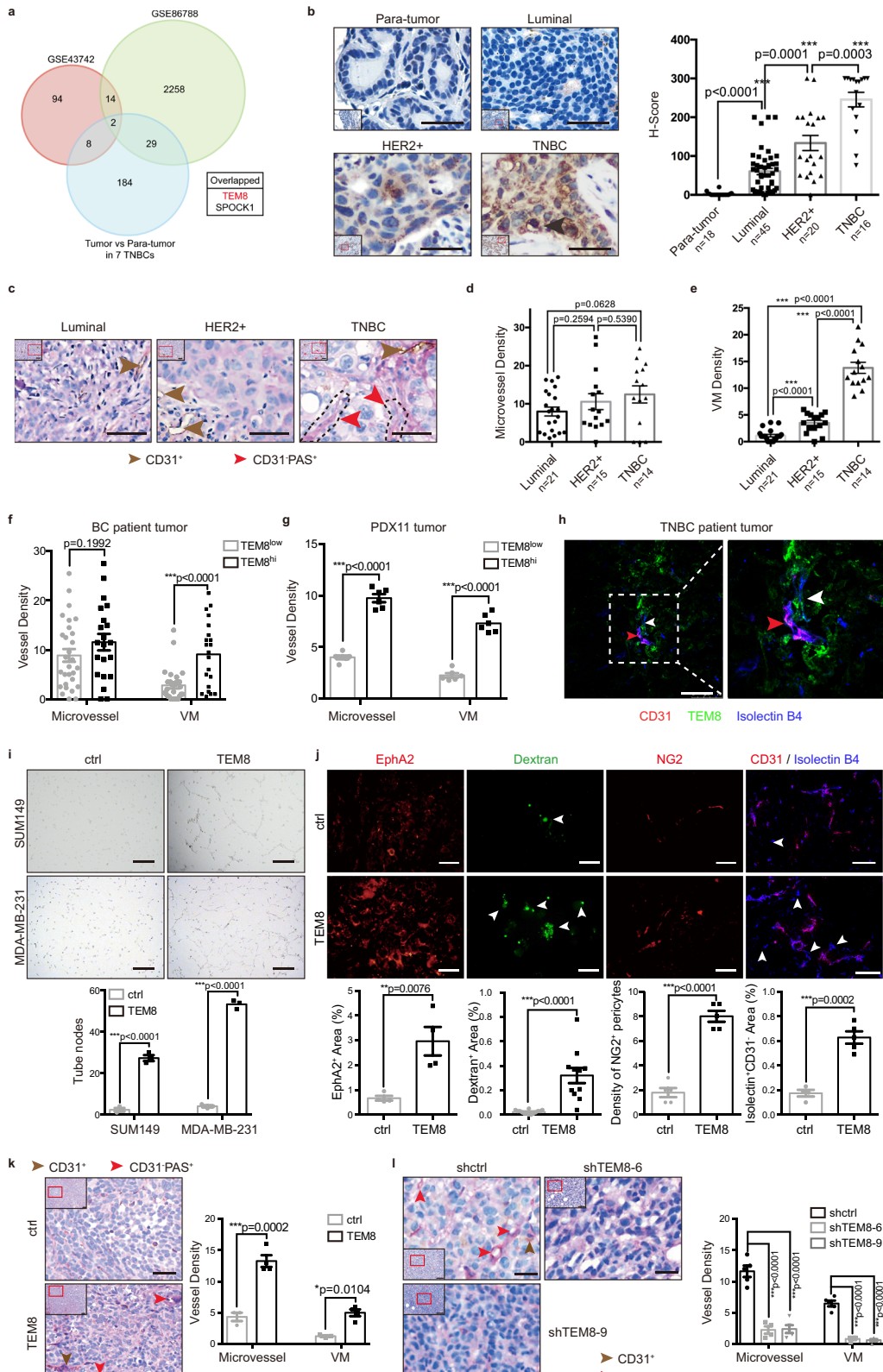

markers such as EphA2 and VE-cadherin in both TNBC patient samples and cell lines (Supplementary Fig. 1g, h). Xenograft experiments supported the effect of TEM8 overexpression on increasing EphA2 expression, dextran leakage, pericyte density, and tumor vessel density (Fig. 1j, k), whereas TEM8 knockdown decreased tumor vessel density (Fig. 1l). Taken together, the results suggested that TEM8 promoted VM in breast tumor cells.

**High TEM8-expressing BTICs had increased VM-forming capacity.** The VM-forming capacity of tumor cells was closely correlated with stem-like property[21]. Based on our results that TEM8 was preferentially expressed in ALDH+ BTICs and that TEM8 expression was positively correlated with five ALDH family members (ALDH1A1/1A2/1A3/1B1/2) (Supplementary Fig. 4a–c), which were stem cell markers in BC and responsible

**Fig. 1 TEM8 promoted vasculogenic mimicry of TNBC cells. a** Overlap analysis of the upregulated genes in TNBC tumors (tumor v.s. para-tumor >2 fold) and two angiogenic gene clusters. **b** Immunohistochemistry analyses of TEM8 expression in BC patient para-tumor and tumor tissues. Representative images (left) and the graph of H Scores (right, mean ± SEM) were shown. Black arrowheads indicated TEM8 highly expressed cells. Scale bar, 20 μm. **c–e** Tumor-associated neovessel analyses of clinical BC tissues. Representative images were shown in (**c**). Quantification of MVD (**d**) and VMD (**e**) in different subtypes. The graph presented a mean ± SEM of 50 biological independent samples. Scale bar, 20 μm. **f** Divided all tumor samples in (**c**) according to TEM8 expression in (**b**), MVD and VMD were then quantified. The graph presented a mean ± SEM of 50 biological independent samples. TEM8[low], low TEM8-expressing tumors (H Score <150); TEM8[hi], high TEM8-expressing tumors (H Score ≥150). **g** Tumor-associated neovessel analyses of TEM8[hi] and TEM8[low] PDX11 cell-derived tumors (five NOD/SCID mice per group, $10^5$ cells/site). Quantification of MVD and VMD were shown (mean ± SEM). **h** Images of CD31[+] endothelial blood vessels and CD31[-]TEM8[+] non-endothelial blood vessels in TNBC patient tumor tissue. Red arrowhead, CD31[+] vessels; White arrowhead, CD31[-]TEM8[+] vessels. Scale bar, 75 μm. **i** In vitro tube formation analysis of TEM8-overexpressing SUM149 and MDA-MB-231 cells. The graph presented a mean ± SEM of three biological independent experiments. Scale bar, 100 μm. **j** Images and comparisons of EphA2 expression, dextran leakage, NG2[+] pericyte density, and non-endothelial blood vessel density in TEM8-overexpressing MDA-MB-231 (MDA-MB-231-TEM8) or control (MDA-MB-231-ctrl) cell-derived tumors (five nude mice per group, $10^6$ cells/site). Data were presented as mean ± SEM. Scale bar, 100 μm. **k** Tumor-associated neovessel analyses of MDA-MB-231-TEM8 cell-derived tumors (three nude mice per group, $10^6$ cells/site). Quantification of MVD and VMD were shown (mean ± SEM). Scale bar, 20 μm. **l** Tumor-associated neovessel analyses of TEM8-knockdown MDA-MB-231[lung] (MDA-MB-231[lung]-shTEM8) cell-derived tumors (five nude mice per group, $10^6$ cells/site). Quantification of MVD and VMD were shown (mean ± SEM). Scale bar, 20 μm. Source data are provided as a Source Data file.

for ALDH enzyme activity in TNBC[13,22,23], we evaluated the expression of TEM8 in ALDH[+] BTICs. The results showed that both TEM8 mRNA and protein expression were upregulated in ALDH[+] BTICs (Fig. 2a, b). Consistently, TEM8 knockdown significantly decreased the ALDH[+] BTIC population (Fig. 2c) and the expression of stem cell factors (Supplementary Fig. 4d), which were significantly increased in TEM8-overexpressing cells (Figs. 2d, S4e). Similar results were observed in xenograft tumors (Supplementary Fig. 4f, g). Next, we assessed the in vitro self-renewal ability of TNBC cells with different expression levels of TEM8, and found that the mammosphere formation efficiency was significantly suppressed in TEM8-knockdown cells (Fig. 2e, f) and TEM8[low] cells (Fig. 2i), but was significantly increased in TEM8-overexpressing cells (Fig. 2g, h) and TEM8[hi] cells (Fig. 2i). These findings indicated that the in vitro self-renewal ability of TNBC cells was positively correlated with TEM8 expression.

To further examine the tumor-initiating ability, TEM8[hi] and TEM8[low] cells were isolated from a TNBC PDX and engrafted into the 4th mammary fat pads of nonobese diabetic (NOD)/ severe combined immunodeficiency (SCID) mice at a limiting dilution. The TIC frequency was determined as previously[24]. TEM8[hi] cells formed tumors (1/8305 TIC frequency) more potently than TEM8[low] cells (1/69592 TIC frequency) (Fig. 2j), and showed significantly increased tumor growth ability (Supplementary Fig. 4h) and enrichment of ALDH[+] BTICs (Supplementary Fig. 4i).

BC cells were divided into four groups according to TEM8 and ALDH levels, and interestingly, only the TEM8[hi]ALDH[+] BTICs had marked VM capacity (Fig. 2k). Therefore, we speculated that high expression of TEM8 was an indicator of VM-forming TICs. To verify this hypothesis, TEM8[low]ALDH[−], TEM8[low]ALDH[+], TEM8[hi]ALDH[−], and TEM8[hi]ALDH[+] tumor cells were isolated from another TNBC PDX and engrafted into NOD/SCID mice at a limiting dilution. TEM8[hi]ALDH[+] cells showed the strongest tumor growth ability (Fig. 2l, m) and the highest TIC frequency (Fig. 2n). The ALDH[+] BTIC population was further enriched in TEM8[hi]ALDH[+] cell-derived tumors (Fig. 2o). Tumor vessel staining indicated that both the TEM8[hi]ALDH[−] group and TEM8[low]ALDH[+] group showed increased tumor VM density. However, tumor VM density was significantly higher in the TEM8[hi]ALDH[+] group (Fig. 2p). Similar results were observed in TEM8[hi]ALDH[+] MDA-MB-231 xenograft experiments (Supplementary Fig. 4j–n). Collectively, these results indicated that TEM8[hi]ALDH[+] TNBC cells were enriched in a population of BTICs with simultaneous strong tumorigenesis and neovasculogenesis abilities.

**TEM8 increased active RhoC level by recruiting GNAS.** To determine whether the effect of TEM8 on promoting VM was mediated by ligand binding, the truncates with intracellular domain or extracellular domain-deleted (both retained transmembrane region) were stably transfected into MDA-MB-231 cells. The results showed that deletion of the intracellular domain reversed the increase of VM (Supplementary Fig. 5a) and mammosphere formation (Supplementary Fig. 5b) induced by the full-length TEM8 protein. On the other hand, deletion of the extracellular domain did not affect the stem-like properties and VM-forming capacity of TEM8 (Supplementary Fig. 5c, d), indicating that the VM-promoting function of TEM8 was dependent on the intracellular domain.

Given that the VM-promoting function of TEM8 is dependent on its intracellular domain, the underlying mechanisms were explored by co-immunoprecipitation (co-IP)/mass spectrometry analysis (Fig. 3a, b). The previously reported TEM8-interacting proteins were not detected in TNBC cells (Supplementary Table 2). Therefore, we focused on a small GTPase called Ras homolog family member C (RhoC), which regulates cancer stemness[25–27], considering that a band of 20 kDa was detected by silver staining. The direct interaction between TEM8 and RhoC was confirmed by co-IP assay (Fig. 3c) and especially in vitro co-IP assay (Fig. 3d). Because the activity of Rho proteins depends on the transition between an active GTP-bound and an inactive GDP-bound state to transduce signals, we demonstrated that TEM8 binding increased the level of active GTP-bound RhoC. Consistently, deletion of the intracellular domain of TEM8 failed to increase active RhoC (Fig. 3e), whereas deletion of the extracellular domain had no effect on active RhoC (Supplementary Fig. 5e), further supporting our conclusion above. The effects of TEM8 on increasing VM (Fig. 3f) and mammosphere formation (Fig. 3g) were reversed by RhoC knockdown, supporting the critical role of RhoC in mediating the malignant functions of TEM8.

Analysis of the TEM8 protein domains indicated that TEM8 could not directly activate GTPase. However, we noticed that the potential TEM8-interacting protein guanine nucleotide-binding protein G(s) subunit alpha isoforms short (GNAS) identified by co-IP/mass spectrometry was also highly expressed in TNBC (Fig. 3h) and could significantly influence RhoC activity. GNAS knockdown reversed TEM8-induced increase of active RhoC (Fig. 3i and Supplementary Fig. 6b), whereas GNAS overexpression further enhanced RhoC activation (Fig. 3j and Supplementary Fig. 6d). TEM8, RhoC, and GNAS were observed to be co-localized in TNBC cells (Fig. 3k). The direct interaction

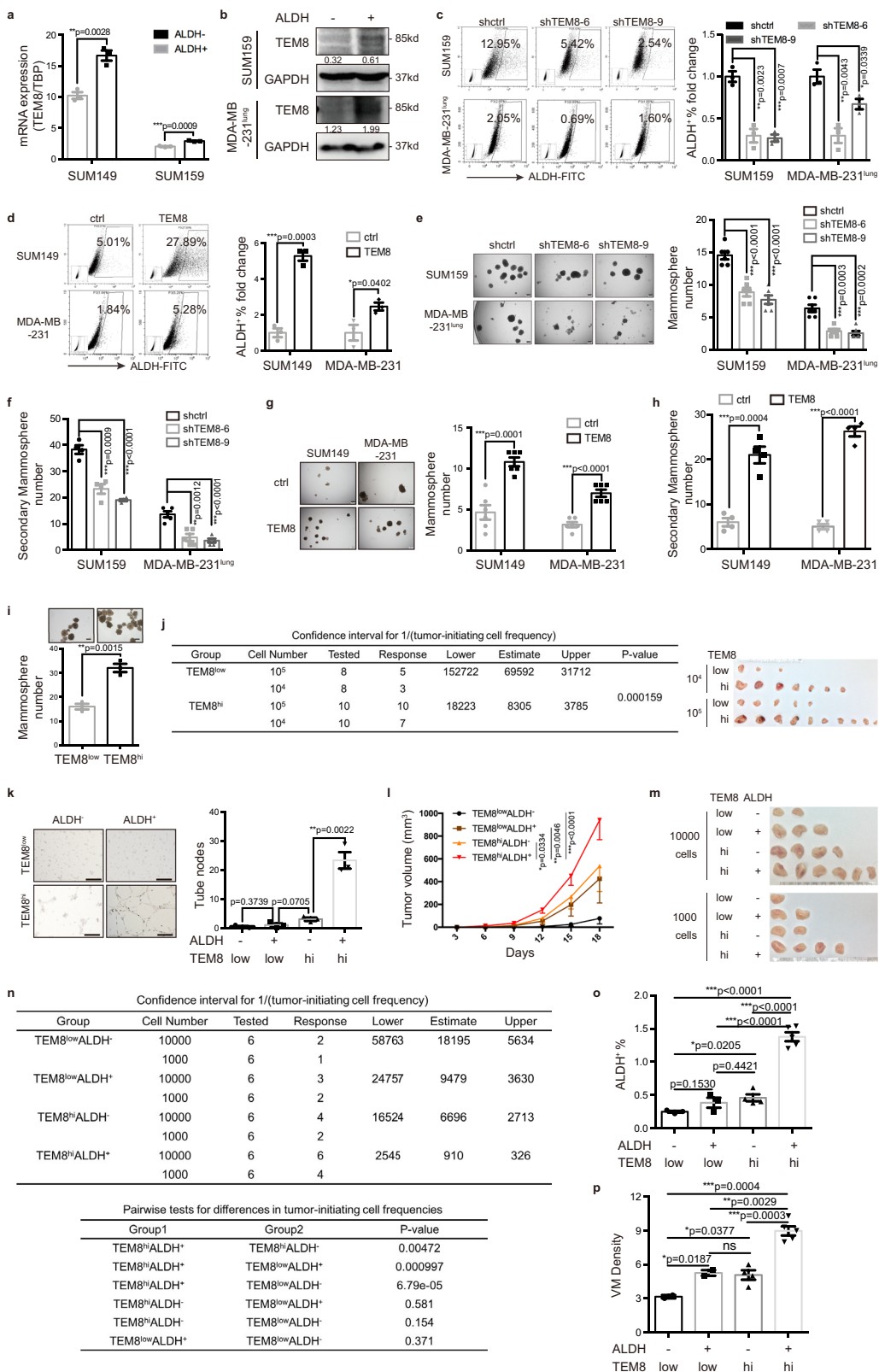

between TEM8 and GNAS was confirmed in an in vitro co-IP system (Fig. 3l), as well as the direct interaction between RhoC and GNAS (Fig. 3m). The presence of TEM8 enhanced the interaction between RhoC and GNAS (Supplementary Fig. 6e). Furthermore, we found that GNAS bound to and increased the active RhoC level via its GTP binding site (Fig. 3n and Supplementary Fig. 6f). The RhoGAP family protein ARHGAP21

was reported to catalyze the conversion of active GTP-RhoC to inactive GDP-RhoC[28]. We found that GNAS overexpression could reverse RhoC inactivation caused by ARHGAP21 (Fig. 3o). Furthermore, GNAS failed to interact with RhoC protein when the ARHGAP21 binding site of RhoC was deleted (Fig. 3p and Supplementary Fig. 6g). Together, these results suggested that GNAS bound to RhoC GAP binding site via its GTP binding site

**Fig. 2 High TEM8-expressing BTICs had increased VM-forming capacity. a, b** mRNA (**a**) and protein (**b**) expression of TEM8 in FACS sorted ALDH+ and ALDH− cells. The graph presented a mean ± SEM of three biological independent experiments. **c, d** ALDH activity detected by ALDEFLUOR assay in TEM8-knockdown cells (**c**) and TEM8-overexpressing cells (**d**). The graph presented a mean ± SEM of three biological independent experiments. **e, f** Self-renewal ability determined by primary mammosphere formation (**e**, $n = 6$ biological independent experiments) and secondary mammosphere formation (**f**, $n = 4$ biological independent experiments for SUM159 and $n = 5$ for MDA-MB-231$^{lung}$) in TEM8-knockdown cells. The graph presented a mean ± SEM. Scale bar, 200 μm. **g, h** Self-renewal ability determined by primary mammosphere formation (**g**, $n = 6$ biological independent experiments) and secondary mammosphere formation (**h**, $n = 4$ biological independent experiments) in TEM8-overexpressed cells. The graph presented a mean ± SEM. Scale bar, 200 μm. **i** Self-renewal ability determined by mammosphere formation in FACS sorted MDA-MB-231$^{lung}$ cells. The graph presented a mean ± SEM of three biological independent experiments. Scale bar, 200 μm. **j** TEM8$^{hi}$ and TEM8$^{low}$ cells sorted from TNBC PDX11 were engrafted into mammary gland fat pads of NOD/SCID mice at limiting dilution. Frequency of TIC was calculated based on the positive tumor sites per group. **k** In vitro tube formation analysis of FACS sorted MDA-MB-231$^{lung}$ cells with different expression level of TEM8 and activity of ALDH. The graph presented a mean ± SEM of three biological independent experiments. Scale bar, 100 μm. **l–p** Cell groups as indicated were sorted from TNBC PDX15 and engrafted into mammary gland fat pads of NOD/SCID mice at limiting dilution. Tumor growth curves (10000 cells/site) (**l**) and tumor images (**m**) were shown. TIC frequency was calculated based on the positive tumor sites per group (**n**). Tumor cell ALDH activity was determined by ALDEFLUOR assay and the bar plot (mean ± SEM) was shown (**o**). Tumor vasculogenic mimicry was stained and the bar plot (mean ± SEM) of VMD was shown (**p**). Source data are provided as a Source Data file.

and thus prevented the canonical interaction between RhoC and ARHGAP21, leading to the activation of RhoC signaling.

**RhoC/ROCK1/SMAD5 signaling pathway mediated TEM8-induced VM.** Rho proteins interact with and activate the Rho-associated protein kinase (ROCK1)[29]. Though ROCK1 coordinated multiple cellular functions, its role in BC and especially in BTICs has not been explored in detail[30]. We, therefore, investigated whether ROCK1 was involved in TEM8-induced TNBC progression. First, we confirmed the interaction between RhoC and ROCK1 (Fig. 4a). Inhibition of ROCK1 with the ROCK1-specific inhibitor (ROCK1i) Y-27632 or shRNA transfection reversed the effects of TEM8 on tumor cell VM (Fig. 4b, c), suggesting that ROCK1 was involved in the function of TEM8.

To identify the predominant downstream effectors of ROCK1 induced by TEM8, a phospho-kinase array was used in TEM8-overexpressing cells to identify alterations of multiple downstream signals, and the results were confirmed by western blotting. TEM8 overexpression did not result in activation of the Wnt pathway (Supplementary Fig. 7a–c), whereas it led to activation of the SMAD5 pathway (Fig. 4d, Supplementary Fig. 7d, e). Although studies showed that the Rho/ROCK pathway was necessary for activation of SMAD2/3 in MCF10CA1h cells[31], we did not observe the activation of SMAD proteins except for SMAD5 in TEM8-overexpressing TNBC cell lines. We speculated that the RhoC/ROCK1 pathway activated by TEM8 could activate SMAD5, which was involved in the stemness of tumor cells and tumor angiogenesis[32–34]. To verify this effect, TEM8-overexpressing cells were treated with RhoC shRNA or the ROCK1i. Both RhoC knockdown (Fig. 4e) and ROCK1i (Fig. 4f) blocked the activation of SMAD5. To confirm the effect of ROCK1 on activating SMAD5, an in vitro phosphorylation assay was performed using purified ROCK1$^{kinase}$ protein and SMAD5 protein. The results showed that SMAD5 was directly phosphorylated at Ser463/465 by ROCK1$^{kinase}$ (Fig. 4g). These data indicated that the activation of SMAD5 induced by TEM8 was mediated directly by the RhoC/ROCK1 pathway.

To evaluate the effect of SMAD5 on the cellular functions of TEM8, SMAD5 was knockdown in TEM8-overexpressing cells. In vitro results showed that knockdown of SMAD5 reversed the TEM8-induced VM (Fig. 4h) and mammosphere formation (Fig. 4i). The in vivo xenograft experiments also showed that SMAD5 knockdown suppressed TEM8-enhanced tumor growth (Fig. 4j), and the enrichment of BTICs (Fig. 4k), as well as the increase in tumor VM density (Fig. 4l). These results demonstrated that TEM8-activated SMAD5 played an important role in promoting BTIC enrichment and tumor VM formation.

**ROCK1i suppressed TEM8 effects on VM and tumorigenesis.** Next, we evaluated the therapeutic effect of blocking TEM8/RhoC/ROCK1/SMAD5 pathway using the ROCK1 inhibitor Y-27632. TEM8-overexpressing MDA-MB-231 cells were engrafted into mammary fat pads of immunodeficient mice, which were treated with Y-27632. The results showed that Y-27632 significantly suppressed in vivo tumor growth accelerated by TEM8 (Fig. 5a and b), as well as BTIC enrichment (Fig. 5c) and tumor VM density (Fig. 5d). Considering that chemotherapy might trigger TIC enrichment and tumor vessel anomalies leading to drug-resistance[35,36], combination treatment with TEM8 blocking agents and chemotherapy might serve as a therapeutic strategy in TNBC. We, therefore, utilized a TNBC PDX with high TEM8 expression to evaluate the therapeutic effect of Y-27632 combined with docetaxel treatment. Y-27632 treatment alone inhibited xenograft tumor growth and enhanced the inhibitory effect of docetaxel treatment (Fig. 5e). Y-27632 combined with docetaxel showed not only a significant decrease of tumor VM density (Fig. 5f), but also a significant decrease in the capacity to initiate secondary tumors (Fig. 5g). These results indicated that inhibition of ROCK1 significantly suppressed the tumorigenesis and neo-vasculogenesis of TEM8-expressing TNBC in vivo, and that the combination of ROCK1 inhibition with chemotherapy could achieve a better therapeutic outcome.

**TEM8 was ubiquitylated by ERα trans-activated ASB10.** Though TEM8 protein expression was significantly higher in TNBC, the mRNA expression of TEM8 was not higher in TNBC than in other BC subtypes (Fig. 6a). Furthermore, high TEM8 expression predicted a poorer clinical outcome only in ER-negative patients (Fig. 6b). We speculated that the difference between the mRNA and protein expression of TEM8 in BC subtypes might originate from post-translational modifications. Treatment of BC cells with the protein synthesis inhibitor cycloheximide (CHX) led to the rapid degradation of the TEM8 protein (Supplementary Fig. 8a). Moreover, TEM8 accumulated significantly in BC cells treated with the proteasome inhibitor MG132 (Supplementary Fig. 8b). The results of ubiquitylation assay showed that TEM8 was ubiquitylated in BC cells (Supplementary Fig. 8c), implying that the TEM8 protein was regulated by the ubiquitin-proteasome system.

To elucidate the regulatory mechanisms underlying TEM8 stability, a Yeast-Two-Hybrid system was used to screen E3 ligases. Six potential E3 ligases were identified (Fig. 6c). Transfection of shRNAs against the identified E3 ligases into MDA-MB-231-TEM8 cells showed that degradation of TEM8 was decelerated significantly by ankyrin repeat and SOCS box protein 10 (ASB10) knockdown (Fig. 6d), whereas it was

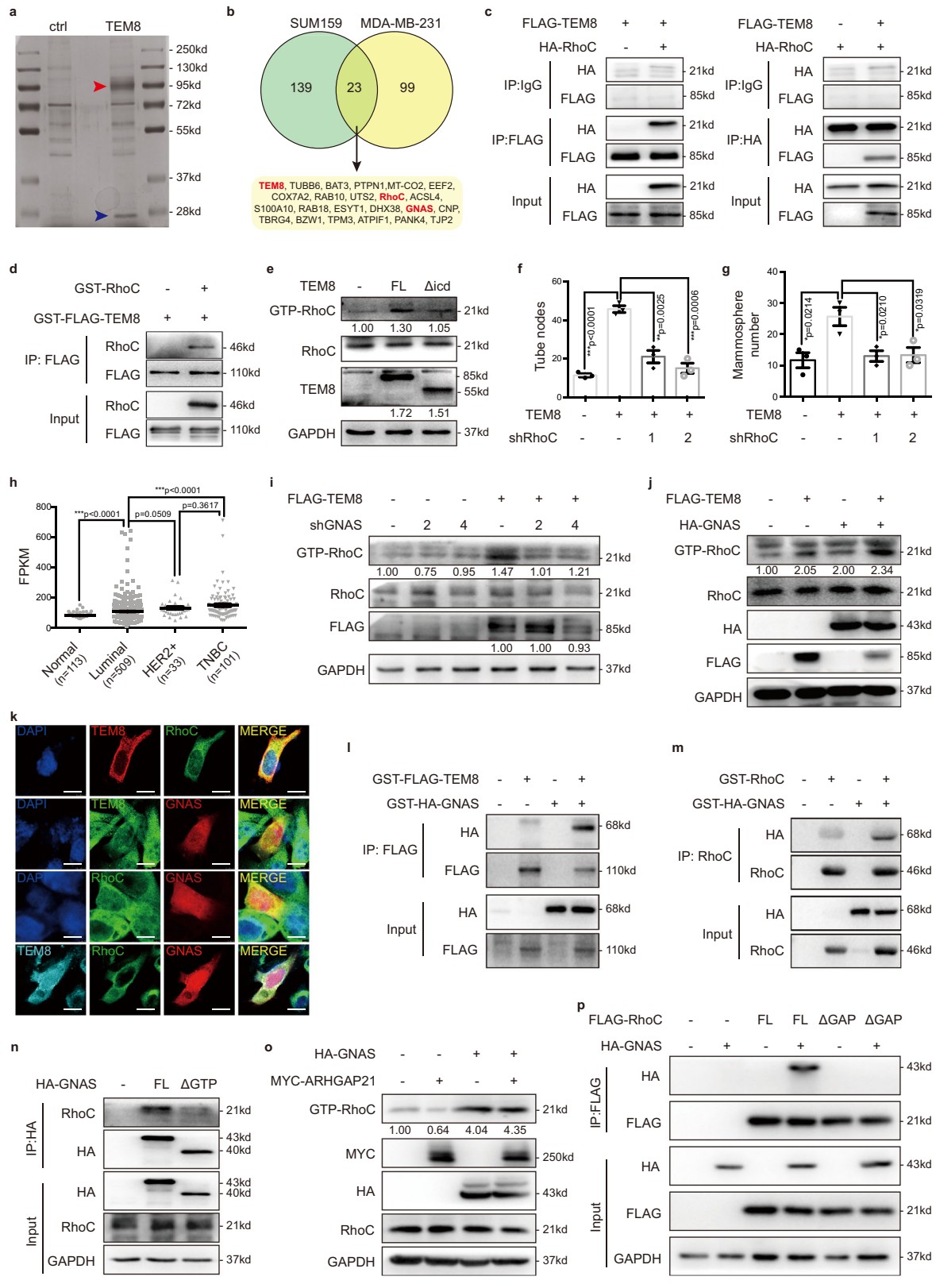

accelerated significantly by ASB10 overexpression (Supplementary Fig. 8e). The results of ubiquitylation assay showed that ASB10 knockdown decreased the ubiquitylation level of TEM8 (Fig. 6e), whereas ASB10 overexpression increased it (Fig. 6f). Moreover, ASB10 knockdown potentiated the effect of TEM8 on increasing cell invasion (Fig. 6g) and VM formation (Fig. 6h). Collectively, these data supported a negative regulatory role of ASB10 in TEM8 protein stability and cellular functions.

Assessment of ASB10 expression in BC patient tissues showed that ASB10 expression was highest in luminal breast tumor

**Fig. 3 TEM8 increased active RhoC level by recruiting GNAS. a** Representative image of silver staining after co-IP in MDA-MB-231-TEM8 cells. Red arrowhead, band with 90kd; Blue arrowhead, band with 20 kd. **b** Mass spectrometry analysis of the co-IP samples (TEM8-overexpressing and ctrl) indicated proteins possibly interacted with TEM8. **c** Co-IP assay analyzed the interaction between TEM8 and RhoC in MDA-MB-231 cells. **d** In vitro co-IP assay analyzed the interaction between TEM8 and RhoC. **e** Full length TEM8 protein (FL) and intracellular domain-deleted truncate (Δicd) were stably transfected into MDA-MB-231 cells. The active RhoC was determined by the RhoGTPase activity assay and quantitated to total RhoC. **f, g** RhoC/scrambled shRNA was transfected into MDA-MB-231-TEM8 cells. The tube formation (**f**) and mammosphere formation ability (**g**) were determined. The graph presented a mean ± SEM of three biological independent experiments. **h** Expression of *GNAS* in normal breast and breast cancer based on TCGA database. The graph presented a mean ± SEM. **i, j** GNAS/scrambled shRNA (**i**) or overexpression (**j**) was transfected into 293T-TEM8 cells. The active RhoC was then determined and quantitated to total RhoC. **k** Immunofluorescence staining analyzed the distribution of TEM8, GNAS and RhoC in SUM159 cells. Scale bar, 10 μm. **l** In vitro co-IP assay analyzed the interaction between TEM8 and GNAS. **m** In vitro co-IP assay analyzed the interaction between RhoC and GNAS. **n** Full length GNAS protein (FL) and GTP binding site (aa357-374)-deleted truncate (ΔGTP) were transfected into 293 T cells. The interaction between GNAS and RhoC was analyzed by the co-IP assay. **o** ARHGAP21 and GNAS were transfected into 293 T cells as indicated. The active RhoC was then determined and quantitated to total RhoC. **p** Full length RhoC protein (FL) and GAP binding site (aa34-39)-deleted truncate (ΔGAP) were co-transfected into 293 T cells. The interaction between GNAS and RhoC was analyzed by co-IP assay. Source data are provided as a Source Data file.

tissues, and lowest or absent in HER2$^+$ or TNBC breast tumor tissues (Fig. 6i, Supplementary Fig. 8f, g). ASB10 expression in BC cell lines was consistent with the IHC results (Supplementary Fig. 8h). TCGA analysis showed that ASB10 mRNA levels were higher in luminal BCs, though the difference did not reach statistical significance, which was probably due to insufficient sample data (Supplementary Fig. 8i). Co-staining of ASB10 with hormone receptors showed that most ASB10$^+$ cells were located within the ERα$^+$ tumor cell region (Figs. 6j, S8j, k), suggesting a positive correlation between ASB10 and ERα.

There were two predicted binding sites of ERα in the promoter region of ASB10 (Supplementary Fig. 8l), which led us to speculate that ERα might transcriptionally regulate the expression of ASB10. To support this hypothesis, we first examined the expression of ASB10 in ERα-overexpressing tumor cells, which showed that ERα overexpression significantly upregulated ASB10 both at the mRNA level (Fig. 6k) and the protein level (Fig. 6l). A ChIP-qPCR assay confirmed the binding of ERα to the promoter sequence of ASB10 (Fig. 6m). Then, the promoter region sequence was inserted into the luciferase reporter vector and a dual-luciferase reporter assay was performed. Co-transfection of ERα significantly increased fluorescence intensity (Fig. 6n), whereas transfection of ERα with deleted binding sites decreased fluorescence intensity (Supplementary Fig. 8m). Furthermore, ASB10 knockdown in ERα$^+$ MCF-7 cells significantly upregulated the TEM8 protein (Supplementary Fig. 8n). Taken together, these results indicated that ERα binded to the promoter region of ASB10 and activated the transcription of ASB10, providing a reasonable explanation for the high TEM8 protein level and TEM8-related VM rich phenomenon in TNBC.

## Discussion

TEM8 was upregulated in correlation with vascular growth during embryonic development and increased in the vasculature of multiple tumor types[15,18,37]. TEM8 was widely expressed in breast tumors, and high expression levels of TEM8 were detected in the stroma adjacent to TNBC tumors[38,39]. Here, we showed that TEM8 is overexpressed in TNBC cells, but not in stromal cells; particularly high expression levels of TEM8 were observed in a subset of BTICs, which may confer them stronger tumorigenic properties and VM capacity.

Previous studies showed a correlation between TEM8 and tumor angiogenesis[20]. Here, we provided a deeper understanding of the role of TEM8 in tumor-associated neovasculogenesis. We showed that high TEM8-expressing breast tumors had higher tumor microvessel density and significantly higher VM density, revealing the role of TEM8 in regulating VM in TNBC. VM played a vital role in tumor malignancy, which reflected the

plasticity of aggressive tumor cells[7,40]. The plasticity of TICs has been associated with VM capacity in multiple tumor types, including BC[6]. In a previous study, TEM8 was correlated with the stem cell phenotype in BC[19]. Our study identified a role of TEM8 in BTICs, as only high TEM8-expressing BTICs potentiated the VM capacity. These findings suggested a further stratification of BTICs in TNBC based on TEM8 expression, and that high TEM8-expressing VM-forming BTICs could be effective targets for TNBC therapy.

Several proteins could bind to and regulate TEM8, such as the cleaved C5 domain of collagen alpha 3 (VI) (Col6A3), the LDL receptor-related protein LRP6, and the urokinase-type plasminogen activator (uPA)[41–43]. We found that stimulation with uPA or Col6A3 did not significantly activate the TEM8 downstream RhoC/SMAD5 pathway in TNBC cells (Supplementary Fig. 5f and g). The VM-promoting function of TEM8 might not be strongly influenced by known ligands. Our results have revealed that TEM8 increased RhoC signaling through recruiting GNAS. The balance between active form and inactive form of Rho GTPase is generally regulated by Guanine nucleotide exchange factors (GEFs) and GTPase-activating proteins (GAPs), as well as the Rho GDP dissociation inhibitor (RhoGDI). While both TEM8 and GNAS did not directly activate RhoC, our results indicated a non-canonical activation pathway of RhoC that GNAS bound to GAP binding site of RhoC and thus prevented the interaction of RhoC and its RhoGAP, ARHGAP21, therefore leading to the increased activation of RhoC signaling. Similar non-canonical Rho GTPase activating pathway has been reported before[44].

RhoC/ROCK pathway regulated tumor angiogenesis[45]. Here, we identified SMAD5 as a direct target of ROCK1 kinase. The RhoC/ROCK1/SMAD5 axis was responsible for the TEM8-induced VM capacity in TNBC. Inhibition of this pathway with the ROCK1 inhibitor Y-27632 suppressed BTIC enrichment and VM formation, leading to significant inhibition of tumor growth. Consistent with our observation, other Rho-kinase inhibitors such as fasudil have been shown to disturb VM formation[46]. Taken together, the present results suggested that targeting ROCK1 was a therapeutic option against neovasculogenic TICs in TNBC. Combination treatment with a ROCK1 inhibitor and chemotherapy might improve the therapeutic effects.

Previous studies have described the modification of the TEM8 protein. TEM8 was palmitoylated to prevent association with Cbl, which in turn regulated the ubiquitylation of TEM8 during PA binding to control the endocytosis of anthrax toxin in HeLa cells[47]. Here, we identified ASB10 as the protein responsible for TEM8 ubiquitylation in BC. Little is known about the function of ASB10 in tumors. We reported the role of ASB10 as an E3 ligase regulating the stability of TEM8 protein in BC and found that ASB10 was trans-activated by ERα, which explained, at least in

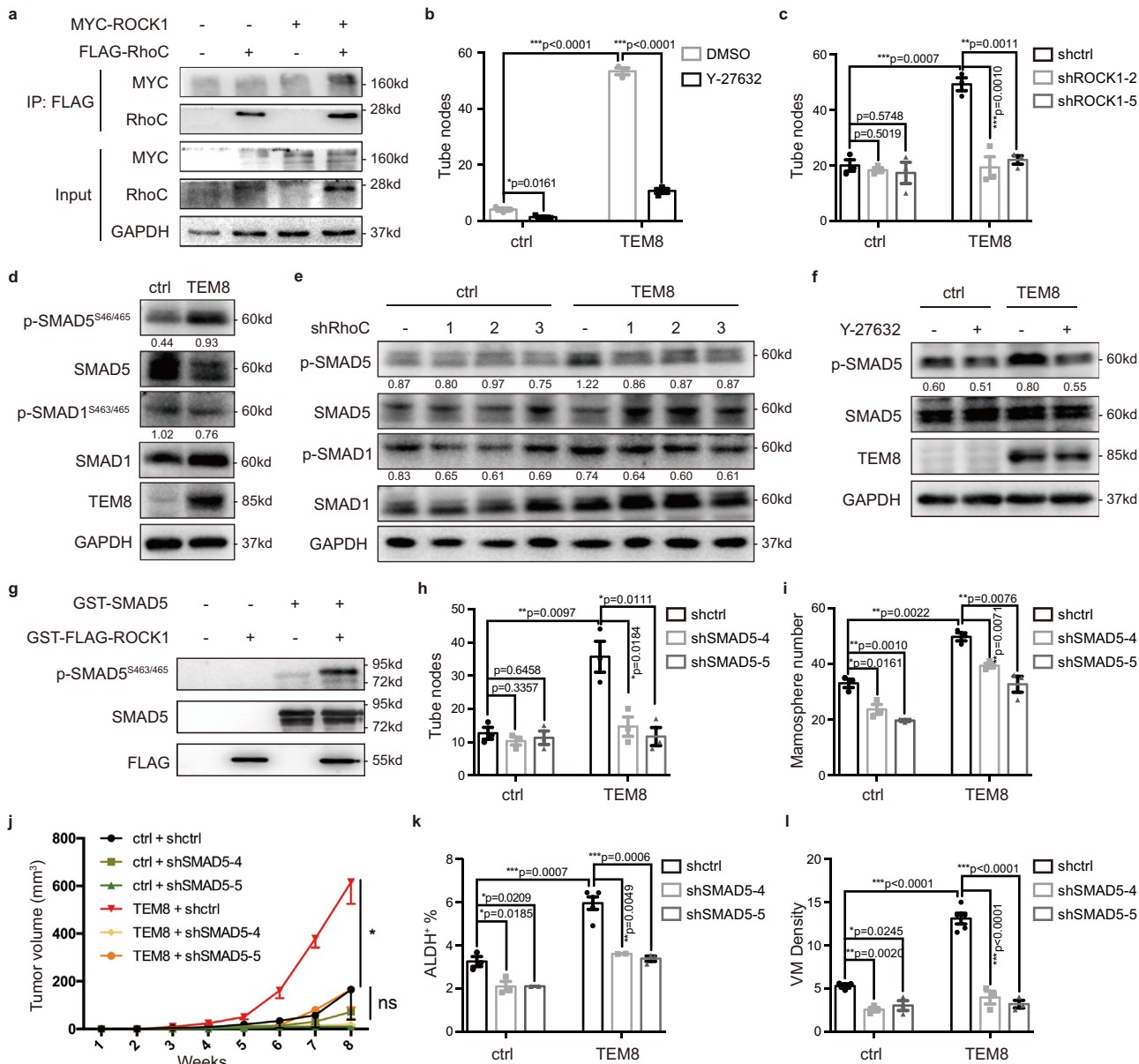

**Fig. 4 RhoC/ROCK1/SMAD5 signaling pathway mediated TEM8-induced VM. a** Co-IP assay analyzed the interaction between RhoC and ROCK1 in MDA-MB-231 cells. **b**, **c** Inhibition of ROCK1 by ROCK1 inhibitor Y-27632 treatment at a concentration of 1 μM for 24 h (**b**) or transfecting ROCK1/ scrambled shRNA (**c**) into MDA-MB-231 cells. Vasculogenic mimicry formation was then determined. The graph presented a mean ± SEM of three biological independent experiments. **d** The expression of SMAD proteins in MDA-MB-231-TEM8 cells analyzed by western blotting. **e**, **f** The expression of SMAD5 protein in MDA-MB-231-TEM8 cells transfected with RhoC/scrambled shRNA (**e**) or treated with Y-27632 at a concentration of 1 μM for 24 h (**f**) was analyzed by western blotting. **g** In vitro phosphorylation assay analyzed the kinase-substrate interaction between ROCK1 kinase domain and SMAD5 protein. **h**, **i** SMAD5/scrambled shRNA was transfected into MDA-MB-231-TEM8 cells. Vasculogenic mimicry (**h**) and mammosphere (**i**) formation were determined. The graph presented a mean ± SEM of three biological independent experiments. **j–l** SMAD5-knockdown MDA-MB-231-TEM8 cells were injected into mammary fat pads (three nude mice per group, 10^6 cells/site). The tumor growth curve was shown (**j**). Tumor cell ALDH activity was determined by ALDEFLUOR assay and the bar plot (mean ± SEM) was shown (**k**). Tumor vasculogenic mimicry were stained and the bar plot (mean ± SEM) was shown (**l**). Source data are provided as a Source Data file.

part, the opposite expression patterns of ASB10 and TEM8 in different BC subtypes.

In summary, our studies demonstrate that TEM8 markes a population of neovasculogenic BTICs in TNBC that play vital roles in VM formation, providing a potential therapeutic strategy for TNBC. We elucidate a potential mechanism regulating TEM8 protein stability by the ERα-trans-activated ASB10. Therefore, the emerging PROTAC technology designed to target TEM8 protein degradation might serve as a therapeutic method to target both

tumor vasculature and TICs in TNBC. The present findings suggest that TEM8 and its downstream effectors are promising targets for the therapeutic inhibition of tumorigenesis and neo-vasculogenesis in TNBC.

## Methods

**Cell culture.** Human breast cancer cell lines SUM149 and SUM159 got from Asterland Bioscience were culture in F12 medium (Gibco) with 5% fatal bovine serum (FBS) (Gibco), 1% streptomycin/penicillin (Beyotime Biotechnology),

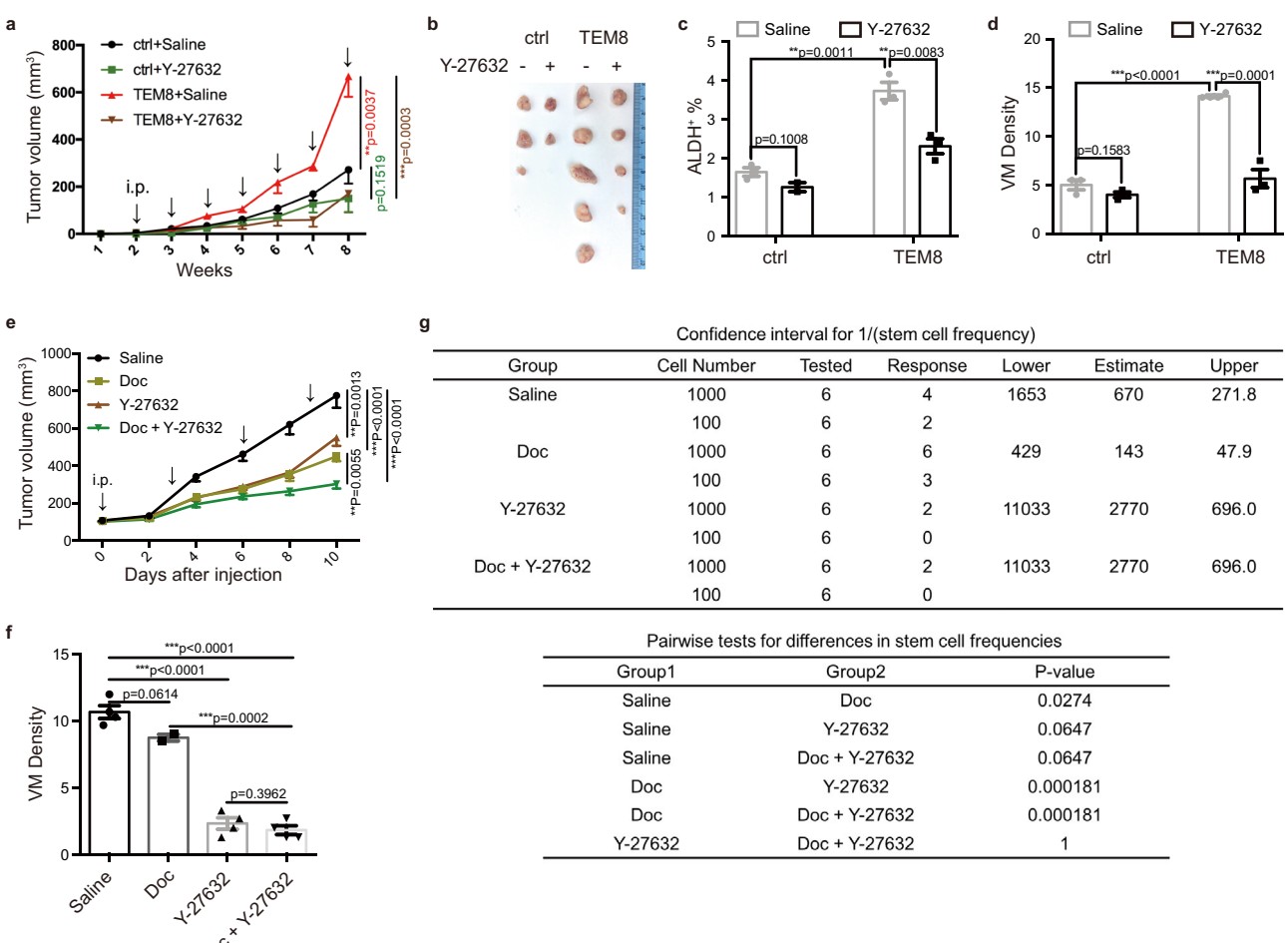

**Fig. 5 ROCK1i suppressed TEM8 effects on VM and tumorigenesis. a–d** The effect of Y-27632 treatment (10 mg/kg, intraperitoneally, weekly) on MDA-MB-231 cell (TEM8-overexpressing and ctrl) tumor growth in nude mice (three mice per group, two sites each mouse, $2 \times 10^6$ cells/site). The tumor growth curve (**a**) and representative image of tumors (**b**) were shown. Time points of treatment were indicated by black arrows. Tumor cell ALDH activity was determined by ALDEFLUOR assay and the bar plot (mean ± SEM) was shown (**c**). Tumor vasculogenic mimicry was stained and the bar plot (mean ± SEM) was shown (**d**). **e, f** The effect of Y-27632 treatment (10 mg/kg, intraperitoneally, every three days) combined with docetaxel therapy (10 mg/kg, intraperitoneally, every three days) on TNBC PDX15 tumor growth in NOD/SCID mice (six mice for each group, two sites each mouse, $10^5$ cells/site). The tumor growth curve was shown (**e**). Time points of treatment were indicated by black arrows. Tumor vasculogenic mimicry was stained and the bar plot (mean ± SEM) was shown (**f**). **g** Tumor cells (H2kd⁻) isolated from Y-27632 and/or docetaxel-treated PDX15 tumor were engrafted in limited dilution to mammary fat pads of NOD/SCID mice (three mice for each group, two sites each mouse, 100 or 1000 cells/site). The stem cell frequency and p value calculation were based on the positive tumor sites. Source data are provided as a Source Data file.

5 mg/ml insulin (Biosharp Life Science) and 1 mg/ml hydrocortisone (Sigma-Aldrich). MDA-MB-231 and MDA-MB-231$^{lung}$ obtained from ATCC. MDA-MB-231 cells were culture in RPMI 1640 medium (Gibco) with 5% FBS and 1% streptomycin/penicillin. MDA-MB-231$^{lung}$ cells were culture in DMEM medium (Gibco) with 5% FBS and 1% streptomycin/penicillin. All the cells were culture in incubators (37 °C, 5% $CO_2$).

**Plasmid construction, lentivirus production, and transfection**. The full length of TEM8 ORF and its truncates with Flag tag were cloned into the lentiviral vector pSIN (Addgene). TEM8 cytoplasmic tail-deleted truncate included extracellular domain, transmembrane domain, and subsequent 9 amino acid residues (WPLCCTVII); TEM8 extracellular domain-deleted truncate included 6 amino acid residues (HCSDGS) and subsequent transmembrane domain, intracellular domain. RhoC, GNAS, and ASB10 with HA tag were cloned into lentiviral vector pLVX (Addgene). The shRNA sequence of TEM8, RhoC, ROCK1, SMAD5, and ASB10 were cloned into lentiviral vector pLKO.1 (Addgene). FLAG-tagged TEM8, HA-tagged GNAS, RHOC, RBD domain of RTKN, SMAD5, and ROCK1 kinase domain were cloned into bacterial expression vector pGEX-6p-1 (Addgene). The plasmid DNA was transduced into 293 T cells to produce a high titer of lentiviruses. The breast tumor cells were then transfected with lentiviruses to establish stable cell lines. The bacterial expression plasmid was transduced into BL21 strain

to produce the expected protein. The primers used for cloning were listed in Supplementary Table 3.

**Total RNA isolation and qRT-PCR**. Total RNA from cells or tissues was extracted by the TRIzol reagent (Takara Bio) and the complementary DNA (cDNA) was obtained from 1 μg RNA using the HiScript II 1st Strand cDNA Synthesis kit (Vazyme Biotech) according to the manufacture's recommendation. The qPCR was performed with AceQ qPCR SYBR Green Master Mix (Vazyme Biotech) in a real-time PCR system (Applied Biosystems) according to the manufacture's recommendation. Primer information was provided in Supplementary Table 4.

**Antibody production and purification**. A recombinant vector pET28a encoding the extracellular domain (ecd) of human TEM8 (aa 33 to 321) protein was transfected into BL21 (DE3) strain. After inducing the expression of the fusion protein, the TEM8-ecd protein was purified from culture supernatant using nickel agarose affinity chromatography. Proteins that were at least 90% pure by coomassie gel staining were used for immunoselection.

After purification, at least 3 mg purified protein was used for further immunization of New Zealand white rabbits, which were immunized with complete Freund's adjuvant and incomplete Freund's adjuvant, respectively. After injecting subcutaneously four times, the immune serum of two groups was collected (RB9075 and RB9076). After preliminary testing in MDA-MB-231-TEM8

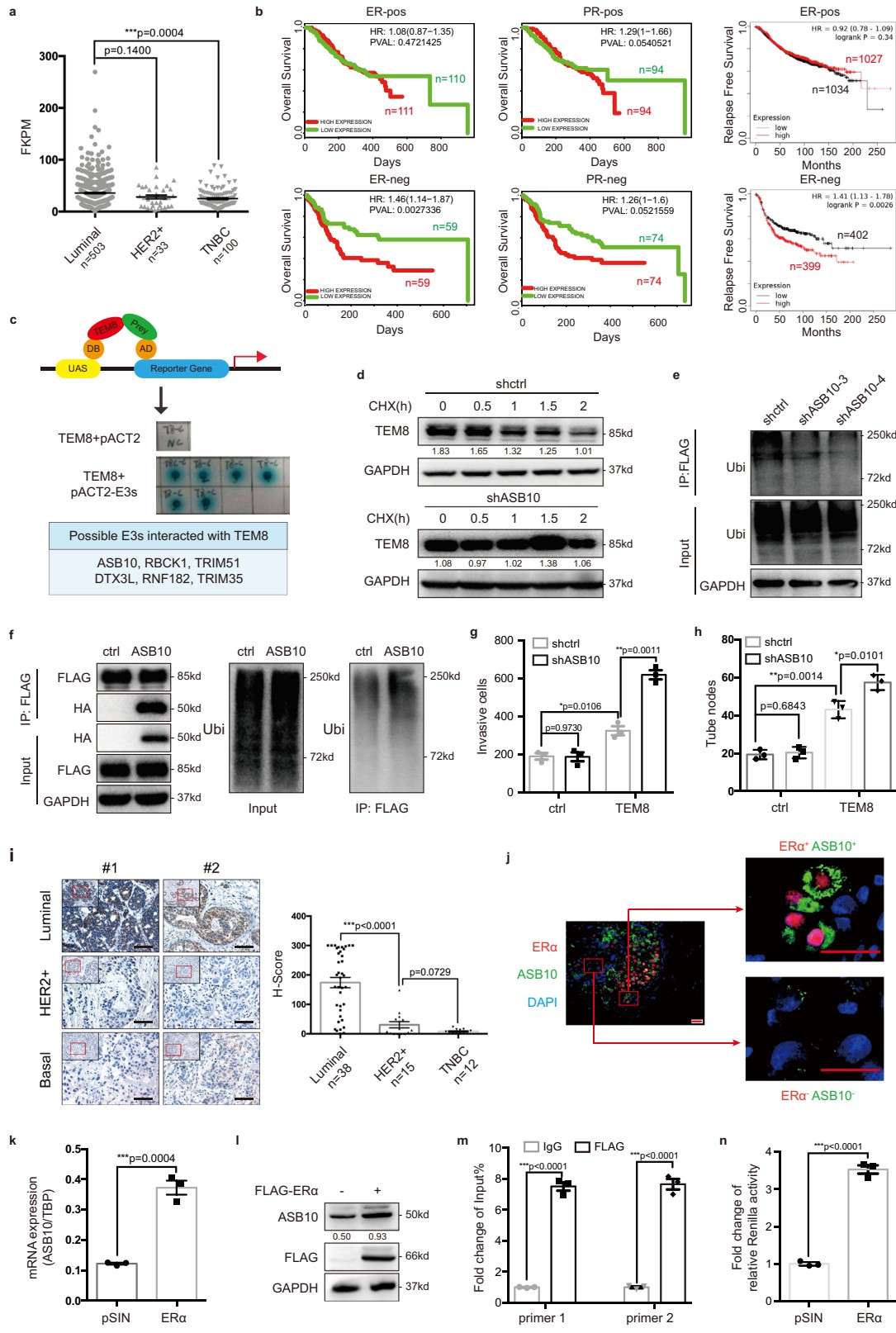

cells by western blotting and immunohistochemistry, RB9075 was selected for further purification using the protein A/G.

**Breast tumor cell isolation and Flow cytometry**. In order to better study the characteristics of breast cancers, we have successfully established a serial of patient-derived xenografts (PDXs) models. Among them, PDX11 and PDX15 were

established from TNBC patient tumor samples and have been stably passaged for two generations in NOD/SCID mice. The subtype of PDX tumors was verified by IHC staining for ER, PR, and HER2 protein.

Xenografted tumors or PDX harvested from mice were minced into small pieces and suspended in collagenase-hyaluronidase digestion solution (StemCell Technologies). The tumor pieces were put into shaker and digested at 37 °C for 1 h. Cell suspension was obtained after filtering with 40 μm filter. The ammonium

**Fig. 6 TEM8 was ubiquitylated by ERα trans-activated ASB10. a** Analysis of the expression of *TEM8* in different subtypes of BC based on TCGA database. Data were presented as mean ± SEM. **b** Overall survival analysis of ER-pos/neg and PR-pos/neg BC patient (GSE22133) based on TEM8 expression. Relapse-free survival analysis of ER-pos/neg BC patient based on TEM8 expression. Cohorts were divided at median of TEM8 expression. **c** E3 ligase screening by Yeast-Two-Hybrid system identified proteins possibly interacting with TEM8. **d–f** ASB10/scrambled shRNA or overexpression (HA-tagged) were stably transfected into MDA-MB-231-TEM8 (FLAG-tagged) cells. TEM8 protein degradation (**d**) and ubiquitylation (**e**, **f**) were determined by western blotting. **g, h** ASB10/scrambled shRNA was stably transfected into MDA-MB-231-TEM8 cells. The cell invasion (**g**) and vasculogenic mimicry formation (**h**) were then analyzed. The graph presented a mean ± SEM of three biological independent experiments. **i** Immunohistochemistry analysis of ASB10 protein expression in clinical BC tissues (*n* = 67). Representative images and the dot plot of *H* Score (mean ± SEM) were shown. Scale bar, 100 μm. **j** Double immunofluorescence staining of ERα and ASB10 in clinical BC tissue (BC106). Scale bar, 50 μm. **k, l** The mRNA (**k**) and protein (**l**) expression of ASB10 in 293T-ERα cells. The graph presented a mean ± SEM of three biological independent experiments. **m** The ChIP-qPCR assay analyzed the binding ability of ERα to the promoter region of ASB10 in 293 T cells. The graph presented a mean ± SEM of three biological independent experiments. **n** The dual luciferase reporter assay analyzed the binding ability of ERα to the promoter region of ASB10 in 293 T cells. The graph presented a mean ± SEM of three biological independent experiments. Source data are provided as a Source Data file.

chloride solution (StemCell Technologies) was used for lysis of erythrocytes for 2 min at RT. The cell aggregates were obtained after centrifuging at 300 g for 5 min and resuspended for subsequent experiments.

For ALDEFLUOR assay (StemCell Technologies), dissociated single cells were suspended in ALDEFLUOR buffer containing ALDEFLUOR substrate BAAA and incubated at 37 °C for 40 min with or without DEAB. For single cells isolated from xenografted tumors, PE-conjugated anti-mouse H2kd antibody (Biolegend, 1:100) was used to discriminate human breast tumor cells from mouse cells. For TEM8 or FLAG-tag staining, single cells were stained with TEM8 antibody RB9075 or anti-FLAG antibody as the primary antibody (dilution at 1:50, the same isotype IgG used as the negative control) and APC- or PE-conjugated donkey anti-rabbit IgG (Jackson ImmunoResearch) as the secondary antibody (dilution at 1:200) on ice for 30 min. Flow cytometry analysis and cell sorting were conducted by MOFLO ASTRIOS (Beckman Coulter) instrument and analyzed by Summit 6.3 software.

**Immunohistochemistry and PAS staining**. All the breast cancer patient tumor tissues and para-tumor tissues were obtained from Shanghai Cancer Hospital affiliated with Fudan University. An informed consent was obtained for all the involved patients, and the study was approved by the institution's ethics committee (Fudan University Shanghai Cancer Center Institutional Review Board, 050432-4-1212B) (Shanghai, China). The tumor tissues from patient or mouse were fixed in formalin and processed for paraffin embedding. The tumor sections were dewaxed and rehydrated in xylene and graded alcohol solutions. The sections were then treated with high-temperature antigen retrieval in citrate buffer (pH 6), and blocked with ready-to-use normal animal serum (Maixin Biotech). The primary antibody was then incubated at 4 °C overnight and secondary antibody at RT for 30 min. After washing, the sections were stained with DAB peroxidase substrate kit (Maixin Biotech) until the desired intensity was achieved. The antibodies used were listed in Supplementary Table 5. The *H* Score was calculated by the formula: percentage of weak staining + 2 × percentage of moderate staining + 3 × percentage of strong staining, giving a range of 0–300. For tumor vessel staining, the Periodic Acid-Schiff (PAS) staining was performed followed CD31 (Abcam and Santa Cruz, 1:50) IHC staining according to the manufacture's protocol (Abcam). Tumor vessel density (including MVD and VMD) calculation by the formula: average of vessel numbers in 3–5 random field (10×).

**RNA Sequencing**. 10 Fresh TNBC tumor tissues and paired para-tumor tissues were obtained from Shanghai Cancer Hospital affiliated with Fudan University. Total RNA was extracted from tissues using TRIzol reagent (Takara Bio). RNA concentration and quality were measured by 2100 Bioanalyzer (Agilent, Santa Clara, CA, USA). RNA-seq libraries were prepared using the NEB Next Ultra Directional RNA Library Prep Kit for Illumina (New England Biolabs, Beverly, MA, USA). The products were sequenced on HiSeq3000 platform (Illumina, San Diego, CA, USA). To analyze data of RNA-seq, FastQC was used for reads quality control. The raw FASTQ files were trimmed for adapter sequences using quart. Then HISAT2 version 2.0.0-beta[48] was used to align reads onto hg19 reference genome with default settings. Gene counts were given by featureCounts Version 2.0.0[49]. The differential gene expression was determined by DEseq using two-fold change, with *p* value <0.05 as threshold.

**Immunofluorescence staining**. Cells of 5×10⁴ seeded in 4-well chamber (Thermo Scientific) were cultured for 2 days and then fixed with cold methanol, membrane perforated with 0.15% Triton X-100 (Sangon Biotech), blocked with normal animal serum (Maixin Biotech), incubated with primary antibody overnight at 4 °C and then secondary antibody for 30 min at RT. The cell nucleus was stained with DAPI (Invitrogen). Images were captured by confocal microscope (LSM710, Zeiss). Antibody information was listed in Supplementary Table 5.

**In vitro tube formation assay**. To assess the VM ability of breast tumor cells, a matrigel-based in vitro tube formation assay was applied. Briefly, 96-well culture plate was coated with 50 μl/well matrigel (BD Biosciences) and incubated at 37 °C for 30 min. The single cells were suspended in complete medium and seeded on pre-coated wells (10⁴ cells/well). The plate was then incubated in 5% CO₂ incubator at 37 °C for 6–10 h. VM structures were photographed with a trinocular microscope (OLYMPUS BX43, Japan) at 4× field. Quantitative evaluation of tube node number was performed by Angiogenesis Analyzer for ImageJ.

**MTT assay**. Breast tumor cells were seeded on 96-well culture plate (100-500 cells/well) and cultured for 3, 5 and 7 days. MTT (Biosharp Life Science) was added to each well (final concentration: 0.5 mg/ml) and cultured in 37 °C for 3 h. Then the supernatant was removed and 100 μl DMSO was added. The optical density (OD) value was measured at 490 nm after the aggregates were dissolved.

**Invasion assay**. The chambers with 8-μm pore size (BD Biosciences) placed into 24-well plate were coated with matrigel and cultured in 37 °C for 3 h. The cells suspended in basal medium were added into pre-coated chambers and the plate was incubated in 5% CO₂ incubator at 37 °C for 36 h. Cells in the lower chamber were fixed with formalin and stained with crystal violet. The images were collected by microscope. Quantitative evaluation of invaded cell number was performed by ImageJ 1.50 g (Java 1.8.0_181) software.

**Mammosphere formation assay**. Breast tumor cells (200 cells/well) were cultured with MammoCult Human Medium kit (StemCell Technologies) supplemented with 1% streptomycin/penicillin, 1 μg/ml hydrocortisone and 4 μg/ml heparin (StemCell Technologies) in an ultra-low attachment 96-well plate (Corning) for 10-14 days. Fresh complete medium was added every four days. The images of mammospheres were collected by microscope.

**Western blotting, protein degradation, and ubiquitylation**. Protein lysates from cells or tissues were extracted using RIPA buffer (Beyotime Biotechnology) and quantified with a BCA kit (Thermo Fisher). the protein lysates were mixed with loading buffer. The protein samples were then separated by SDS-PAGE and transferred onto PVDF membranes (Millipore). The membrane blocked with 5% de-fat milk was incubated with primary antibody overnight at 4 °C and a HRP-conjugated secondary antibody for 30 min at RT. Chemiluminescence was detected using an ImageQuant LAS 4000 mini imaging system (GE) with western HRP Substrate (Millipore). To examine protein degradation, cells were treated with CHX (100ug/ml, MedChemExpress) for several time points before harvested and followed by western blotting. To examine the ubiquitylation level of TEM8, cells were treated with MG132 (20 uM, 2 h, MedChemExpress). Protein lysates extracted by RIPA buffer were incubated with FLAG-beads (Sigma Aldrich) overnight at 4 °C. After washing, proteins were eluted and followed by western blotting detection using an anti-ubiquitin antibody (Santa Cruz, 1:500).

**Immunoprecipitation and mass spectrometry**. Protein lysates extracted from EBC buffer (50 mM Tris, 120 mM NaCl, 0.5% NP40, pH7.5) were incubated with magnetic FLAG-Beads (Sigma-Aldrich) or anti-HA antibody-conjugated protein A/G agarose beads (Thermo Fisher) overnight at 4 °C. After washing with NETN buffer (20 mM Tris, 100 mM NaCl, 0.5% NP40, 1 mM EDTA, pH8.0), the complexes were eluted to follow Western blotting.

For mass spectrometry analysis, cells (MDA-MB-231-ctrl, MDA-MB-231-TEM8, SUM159-ctrl, and SUM159-TEM8) were treated with MG132 (20 uM, 2 h) before harvested. IP samples (on FLAG-beads) were dissolved with 50 mM ammonium bicarbonate solution and digested by trypsin at 37 °C for 16 h. After being centrifuged, the supernatant was freeze-dried and desalted, followed by adding 0.1% formic acid and vortexed to be fully dissolved. The samples were then centrifuged and the supernatant was added to the sample bottle for mass spectrometry (Orbitrap Elite) detection. Search database was Maxquant. In the initial screening, proteins with high LFQ intensity (>10⁶) in TEM8-overexressing

groups but not in ctrl groups (LFQ intensity is 0) were picked out and identified as possible TEM8-interacting proteins. Antibody information used in western blotting and IP were listed in Supplementary Table 5.

**Yeast-Two-Hybrid Screen.** The intracellular domain of human TEM8 ORF was cloned into donor vector pGBKT7 (Clontech Laboratories) to generate the bait plasmid, which expresses TEM8 intracellular domain (icd) protein fused to amino acids 1–147 of the GAL4 DNA binding domain (DNA-BD). The prey vector pACT2 (Clontech Laboratories) containing human cDNA collections in-frame fused to the GAL4 activating domain (AD). Empty pACT2 plasmid was used as a negative prey control. Y2H screening was performed by transforming yeast strain that harbored bait vector pGBKT7-icdTEM8 with prey vector pACT2 carrying human E3 ligase cDNA expression library. The yeast cells were grown on SD-2 plate (deficient in Leu and Trp) for selection of cells containing both bait and prey vectors and then grown on SD-4 plate (deficient in Leu, Trp, His and Ura) for selection of colonies that expressed proteins that potentially interacted with icd-TEM8. The colonies were picked out and grown on another SD-4 plate with X-Gal (Sigma-Aldrich). The colonies that able to grow on the SD-2 or SD-4 plates at 30 °C for three days and shown blue color in X-Gal staining assay for β-galactosidase activity were identified as 'positive'. Images of the colonies on both plates were recorded.

**RhoGTPase activity assay.** Protein lysates were extracted from tumor cells using RIPA buffer (Beyotime Biotechnology). A GST-fusion Rho binding domain (RBD) of the Rho effector protein Rhotekin was purified from BL21 strain using the GST agarose beads, and added into the cell lysates to pull down the GTP-bound form of RhoC overnight at 4 °C. After washing with NETN buffer, the beads were mixed with loading buffer. The amount of GTP-RhoC was determined by subsequent Western blotting using a RhoC specific antibody (CST, 1:1000).

**In vitro kinase assay.** GST-fusion proteins (ROCK1 kinase domain and SMAD5) were purified from BL21 strain and added into the kinase buffer system (50 nM HEPES, 100 mM NaCl, 10 mM $MgSO_4$, 2 mM DTT, 4 mM EDTA, pH 7.2) supplemented with ATP (final concentration: 50 µM). The reaction was conducted at 30 °C for 30 min and then mixed with loading buffer subjected to following Western blotting.

**Human phospho-kinase array.** Cell lysates collected from MDA-MB-231 (ctrl and TEM8-overexpressing) cells were quantified with BCA kit (Thermo Fisher) and then diluted and incubated overnight with the Human Phospho-Kinase Array (R&D Systems, ARY003B) according to the manufacture's protocol. Chemilumi-nescent signal was collected using an ImageQuant LAS 4000 mini imaging system. The pixel density was determined by ImageJ 1.50 g (Java 1.8.0_181) software. The relative kinase protein level was quantified with the reference spot.

**Chromatin immunoprecipitation (ChIP) assay.** In total, 293 T cells were trans-fected with a pSIN vector containing a FLAG-tagged ERα fragment and cultured for two days. The magnetic FLAG-beads (Sigma-Aldrich) were used to pull down the ERα protein and binding DNA. A nonspecific normal IgG was used as negative control. And the purified DNA was used to perform subsequent qRT-PCR detection. We designed two pairs of primers to target different regions of the ASB10 promoter, as listed in Supplementary Table 4.

**Dual luciferase reporter assay.** In total, 293 T cells transfected with pSIN vector containing FLAG-ERα fragment were seeded in 96-well plate and transfected with ASB10 promoter-containing luciferase reporter psiCHECK-2 (Addgene). After transfection for 24 h, the luciferase activity was measured according to the instructions of Dual-Glo Luciferase Assay kit (Promega).

**Survival analysis with online database.** GSE43742 and GSE86788 data for angiogenic gene clusters[50,51]; GSE5327, GSE2603, and GSE2034 data for lung metastasis-free analysis[52–55]; GSE22133 data for overall survival analysis[56,57] were obtained from Gene Expression Omnibus (GEO) of National Center for Bio-technology Information (https://www.ncbi. https://nlm.nih.gov/geo/query/acc.cgi). Kaplan-Meier plotter was used to analyze relapse-free survival of TNBC patients based on TEM8 expression[58]. A total of 1220 BC patients from The Cancer Genome Atlas (https://tcga-data. nci.nih.gov/tcga) were used for gene expression analysis for TEM8, GNAS, and ASB10.

**In vivo tumorigenicity.** Three to four-week-old female Nude mice or NOD/SCID mice were purchased from Vitalriver (Beijing, China) and housed in standard animal cages under a Specific-pathogen-free facility at 23–25 °C on a 12-h light/ dark cycle in the Department of Laboratory Animal Science of Fudan University. All mice experiments were conducted in accordance with standard operating procedures in accordance with the recommendations in the Guide for the Care and Use of Laboratory Animals of Fudan University, and approved by the Committee on the Ethics of Animal Experiments of Fudan University Shanghai Cancer Center

Institutional Review Board (JS-082). For BC cells xenograft experiments, cells were engrafted into mammary fat pads of nude mice. The in vivo treatment of ROCK1 inhibitor started right away xenografts transplanted. For PDX experiments, cells were engrafted into mammary fat pads of NOD/SCID mice. The in vivo treatment of ROCK1 inhibitor combined with docetaxel started when the tumor volume reached 100 mm³. For limited dilution assays, cells isolated from cell lines or PDX were engrafted into mammary fat pads of NOD/SCID mice at limiting dilutions. The tumors were monitored weekly. The tumor volume was calculated as Length × Width × Width/2. Mice were sacrificed when the diameter of tumors reached 10–15 mm.

**In vivo vascular leakage assay.** MDA-MB-231-ctrl and -TEM8 cells ($3 \times 10^6$) were injected into mammary fat pads of nude mice. When the tumor diameter reached a size of 5–10 mm, the tumor vessel leakage was analyzed after intravenous injection of 100 µl FITC-conjugated dextran (70 kDa, 25 mg/ml, Sigma-Aldrich) for 30 min before the mouse sacrifice. Excised tumors were embedded into OTC to prepare for frozen sections. The fluorescence of dextran leakage was observed under a fluorescence microscope.

**Statistic and reproducibility.** All experiments for quantitative analysis and representative images were reproduced with similar results for at least three times. Data sets with normal distribution were analyzed with unpaired Student's two-sided t-tests to compare two groups. Two-sided log-rank (Mentel–Cox) test was used to evaluate the survival analysis. Two-way ANOVA was used to evaluate the difference among three or more groups. Pearson Chi-square test was used to evaluate IHC score levels between different clinicopathological variable groups. Bivariate correlation analysis was performed using the Pearson correlation method. $P < 0.05$ was considered statistically significant. All statistical analyses were per-formed by GraphPad Prism 6.0 software.

**Reporting summary.** Further information on research design is available in the Nature Research Reporting Summary linked to this article.

## Data availability
The publicly available dataset used in this study can be accessed under the GEO accession codes GSE43742, GSE86788, GSE5327, GSE2603, GSE2034, GSE22133. The RNA-seq data profiles used in this study have been deposited in the NCBI under accession code PRJNA739366. All other data supporting the findings of the study are available in this article and its supplementary information files. All other relevant data are available from the corresponding author on request. The source data are provided as a Source Data file. Source data are provided with this paper.

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

## Acknowledgements

This work was supported by the National Key Research and Development Program of China (2020YFA01123002, 2018YFA0507501, 2016YFA0101202), NSFC grants (81772799, 81930075, 81773155, 82073267), Program for Outstanding Medical Academic Leader in Shanghai (2019LJ04), Program of Shanghai Academic/Technology Research Leader (20XD1400700), the innovative research team of high-level local university in Shanghai, the Fudan University Research Foundation (IDH 1340042), the Research Foundation of the Fudan University Shanghai Cancer Center (YJRC1603).

## Author contributions

Study Concept and Design, J.X., L.Z. and S.L.; Acquisition of Data, J.X., X.Y., Q.D., C.Y., D.W., X.Y., X.H. J.D. J.Q. J.T. and R.Z.; Statistical Analysis, G.J. and J.X.; Drafting of the Manuscript, J.X., L.Z. and S.L.; Study Supervision, S.L., L.Z., R.H., X.B., Q.L., Z.S.; Obtained Funding, L.Z., S.L. and X.B.

## Competing interests

The authors declare no competing interests.
