## [Peer Review File · Nature Communications]

Reviewers' Comments:

Reviewer #1:

Remarks to the Author:

The authors of the current study analyse the contribution of breast tumor-initiating cells to aberrant neovasculogenesis in triple negative breast cancer and identify Tumor endothelial marker 8 (TEM8) as one of the most relevant marker involved in vasculogenic mimicry. The authors in this landscape study not only demonstrate the implication of TEM8 in VM but they define the mechanism through activation of ROCK1 to phosphorylate SMAD5, in a cascade essential for promoting stemness and VM capacity of breast cancer cells. They also report the molecular basis for how TEM8 regulate the VM-forming Breast Tumor Initiating Cells and contribute to the neovasculogenesis and tumorigenesis in TNBC the regulatory mechanism in the control of TEM8 proteostasis.

This is an impressive piece of research, however the initial enthusiasm decrease after realising that the role of TEM8 in aberrant vasculogenesis has not been analysed respect to other knowns VM markers and no functional immunofluorescence studies were performed to fully address the role of TEM8 in VM.

Thus, the authors need to perform:

Immunofluorescence images of TEM8 are needed to be shown in the tumor context to give a morphological/visual evidence or their finding with particular focus on the transition between endothelial and non-endothelial tubes, functional analysis of leakage, and presence of pericytes cover as function of TEM8 expression.

Analyse the correlation between known VM markers such as EphA2, FAK, VE-cadherin and TEM8 in TNBC

Reviewer #2:

Remarks to the Author:

In this manuscript the authors demonstrate that in triple negative breast cancer (TNBC), vasculogenic mimicry (VM) is regulated by a pathway involving TEM8- RhoC -ROCK1 and SMAD5. This intern promotes stemness and VM. In general, the experiments are through and well reported and provide mechanistic incites on a physiologically and potentially clinical important pathway. However, several areas need clarification and a more thorough discussion of previous studies are needed.

Specifically:

1) The authors developed a polyclonal antibody to TEM8 which is capable of detecting the protein in tissue sections as well as Western blots and flow studies. Heterogeneity of expression is demonstrated in BC tumor sections. It would strengthen the argument for preferential expression of TEM8 in cancer stem cells (CSC's) if co staining for CSC markers such as ALDH1 or, CD44+/CD24- were shown.

2) Based on studies showing that VM is preserved with TEM8 lacking the extracellular domain, the authors focus exclusively on signaling generated by the intracellular domain. However, by doing this they do not address upstream signaling regulating TEM8 activation. Previous studies have demonstrated that TEM8 functions as a receptor for uPA and mediates uPA stimulated EGFR phosphorylation (Zhang Cell Communication and Signaling 16.62, 2018). This requires further discussion.

3) Utilizing TEM8 and the CSC marker ALDH, the authors define 4 groups and demonstrate that the ALDH+, TEM8+ cells possess the greatest VM capacity in vitro and tumor in initiating potential in vivo. They do not describe the plasticity of these subgroups of cells. Do the cells groups

interconvert when cultured in vitro? If so, what regulates the equilibrium ie what is the effect of hypoxia, an important regulator of angiogenesis.

4) The authors identify a pathway downstream of TEM8 involving RhoC-ROCK1-SMAD5 which intern regulates both stemness and VM. Each of these downstream effectors has previously been linked to cancer stemness. The authors need to cite these studies. Studies showing the effects of RhoC and ROCK -1 inhibitors on CSC are particularly relevant.

5) The authors claim that addition of the ROCK1 inhibitor Y-27632 accentuates the effects of docetaxel chemotherapy. However, Fig5c shows no statically significantly difference in tumor growth between DOC and DOC+ Y-27632.

6) The authors propose that in ER+ breast cancers, Era regulates the expression of ASB10 to degrade TEM8 in these cells. However, the expression of ER+ is heterogenous in ER+ cells and in fact in luminal B breast only a few percent of cells express ER. If this was a cell autonomous mechanism, then the cells in these tumors that lacked Er expression should express TEM8 protein. However, this is not consistent with the data shown. To further support their hypothesis, the authors should demonstrate that ASB10 knockdown, results in increased TEM8 protein expression.

7) The manuscript would benefit from further English editing.

Reviewer #1 (Remarks to the Author):

The authors of the current study analyze the contribution of breast tumor-initiating cells to aberrant neovasculogenesis in triple negative breast cancer and identify Tumor endothelial marker 8 (TEM8) as one of the most relevant marker involved in vasculogenic mimicry. The authors in this landscape study not only demonstrate the implication of TEM8 in VM but they define the mechanism through activation of ROCK1 to phosphorylate SMAD5, in a cascade essential for promoting stemness and VM capacity of breast cancer cells. They also report the molecular basis for how TEM8 regulate the VM-forming Breast Tumor Initiating Cells and contribute to the neovasculogenesis and tumorigenesis in TNBC the regulatory mechanism in the control of TEM8 proteostasis.

This is an impressive piece of research, however the initial enthusiasm decrease after realizing that the role of TEM8 in aberrant vasculogenesis has not been analyzed respect to other known VM markers and no functional immunofluorescence studies were performed to fully address the role of TEM8 in VM.

Thus, the authors need to perform:

1. Immunofluorescence images of TEM8 are needed to be shown in the tumor context to give a morphological/visual evidence or their finding with particular focus on the transition between endothelial and non-endothelial tubes, functional analysis of leakage, and presence of pericytes cover as function of TEM8 expression.

Answer: Thank you very much for your comments and suggestions. In order to show a visual evidence of our finding, we have compared the blood vessel tubes, vascular leakage and pericyte coverage in both TNBC patient tumor tissues and TEM8-overexpressing xenograft tumor tissues via immunofluorescence staining. Immunofluorescence images of blood vessel tubes (shown by isolectin B4 staining) illustrated the presence of CD31⁺ endothelial tubes and CD31⁻TEM8⁺ non-endothelial

tubes inside tumors. In particular, we have observed the transition between CD31⁺ endothelial tubes and CD31⁻TEM8⁺ non-endothelial tubes (Fig. 1h). More severe vascular leakage (shown by 70kDa FITC-dextran perfusion) and higher density of NG2⁺ pericytes were seen in the TEM8-overexpressing xenograft tumors (Fig. 1j).

Fig. 1h Immunofluorescence images of CD31⁺ endothelial and CD31⁻TEM8⁺ non-endothelial blood vessel tubes.

Fig. 1j Images and comparisons of EphA2 expression, dextran leakage, pericyte density and non-endothelial blood vessel density in MDA-MB-231-TEM8 xenografts.

2. Analyze the correlation between known VM markers such as *Epha2*, *FAK*, *VE-cadherin* and *TEM8* in TNBC

Answer: Thanks for your suggestions. The correlation analysis (by bc-GenExMiner v4.5) have shown a positive correlation between VM markers (EphA2 and VE-Cadherin) and TEM8 in TNBC (Supplementary Fig. 1g), which was further confirmed by Western blotting in TEM8-overexpressing TNBC cells (Supplementary Fig. 1h). Immunofluorescence staining also showed higher expression of EphA2 in TEM8-overexpressing xenografts (Fig. 1j).

Supplementary Fig. 1g&h Positive correlation between TEM8 and known VM markers. (g) Correlation analysis of TEM8 and known VM markers (EPHA2 and VE-CADHERIN) using bc-GenExMiner v4.5. (h) Western blotting analysis of the expression of EphA2 and VE-Cadherin in MDA-MB-231-TEM8 cells.

In conclusion, through evaluating the vascular perfusion, leakage, pericyte coverage and the correlation with known VM markers, we think it is justified to declare the important roles of TEM8 in VM formation.

Reviewer #2 (Remarks to the Author):

In this manuscript the authors demonstrate that in triple negative breast cancer (TNBC), vasculogenic mimicry (VM) is regulated by a pathway involving TEM8- RhoC -ROCK1 and SMAD5. This intern promotes stemness and VM. In general, the experiments are through and well reported and provide mechanistic incites on a physiologically and potentially clinical important pathway. However, several areas need clarification and a more thorough discussion of previous studies are needed.

Specifically:

1) The authors developed a polyclonal antibody to TEM8 which is capable of detecting the protein in tissue sections as well as Western blots and flow studies. Heterogeneity of expression is demonstrated in BC tumor sections. It would strengthen the argument for preferential expression of TEM8 in cancer stem cells (CSC's) if co staining for CSC markers such as ALDH1 or, CD44⁺/CD24⁻ were shown.

Answer: Thanks for your thoughtful comments and suggestions. As you suggested, we have co-stained TEM8 and CSC (hereinafter referred to as TIC) markers including ALDH, CD24, and CD44 in TNBC cell line or cells from TNBC PDX tumors via flow cytometry analysis. Our results showed that, ALDH⁺ TICs were contained significantly with higher percentage of TEM8-high cells in all samples (Supplementary Fig. 4a), which was consistent with our previous results (Fig. 2a&b). But we have not found TEM8-high cell enrichment in CD24⁻CD44⁺TICs (Supplementary Fig. 4a).

Supplementary Fig. 4a Flow cytometry analysis of TEM8 expression in TICs and non-TICs of TNBC cell line SUM149 and PDXs. Representative images of SUM149 were shown in the left panel. Statistical results were shown in the right panel.

Fig. 2a&b The mRNA and protein expression of TEM8 in sorted ALDH⁺ and ALDH⁻ cells.

2) Based on studies showing that VM is preserved with TEM8 lacking the extracellular domain, the authors focus exclusively on signaling generated by the intracellular domain. However, by doing this they do not address upstream signaling regulating TEM8 activation. Previous studies have demonstrated that TEM8 functions as a receptor for uPA and mediates uPA stimulated EGFR phosphorylation (Zhang Cell Communication and Signaling 16.62, 2018). This requires further discussion.

Answer: Thank you very much for pointing this out. Our functional studies have strengthened the importance of TEM8 intracellular domain and therefore we mainly focused on the downstream signaling activated by TEM8. Indeed, we have also evaluated the upstream signaling regulating TEM8 activation, such as the known TEM8 ligands urokinase-type plasminogen activator (uPA) and cleaved C5 domain of Collagen α 3(VI) (Col6A3). However, our results showed that, both uPA stimulation and Col6A3 stimulation did not significantly activate TEM8 downstream RhoC/SMAD5 pathway (Supplementary Fig. 5f&g), suggesting that uPA and Col6A3 may not be the major extracellular activators of TEM8 in VM-forming TNBC cells.

Supplementary Fig. 5f&g Western blotting analysis of RhoC and SMAD5 protein activation in MDA-MB-231-TEM8 cells after uPA (f) or Col6A3 (g) stimulation for 1 hour.

3) Utilizing TEM8 and the CSC marker ALDH, the authors define 4 groups and demonstrate that the ALDH⁺, TEM8⁺ cells possess the greatest VM capacity in vitro and tumor in initiating potential in vivo. They do not describe the plasticity of these subgroups of cells. Do the cells groups interconvert when cultured in vitro? If so, what regulates the equilibrium ie what is the effect of hypoxia, an important regulator of angiogenesis.

Answer: Thanks for your questions. Our studies demonstrated that TEM8^{hi}ALDH⁺ cells possessed the strongest stemness properties and TEM8^{low}ALDH⁻ cells possessed the least. It is known that stem cells could differentiate into non-stem cells, while non-stem cells seldom de-differentiate to stem cells without particular external stimulants. We found that TEM8^{hi} cell-derived xenografts contained not only TEM8^{hi} cells but also large amount of TEM8^{low} cells, while TEM8^{low} cell-derived xenografts seldom generated TEM8^{hi} cells (see below Fig. I), suggesting a unidirectional differentiation of TEM8^{hi} cells. To further describe the plasticity of the 4 groups defined by TEM8 and ALDH, we separated these cells from the TNBC cell line SUM159 by fluorescent activated cell sorting (FACS), and cultured them in vitro. After 7 days, we re-examined their TEM8 and ALDH by flow cytometry and our results showed that TEM8^{hi}ALDH⁺ cells could generate a heterogeneous cell population similar to the unsorted parental cells, but TEM8^{low}ALDH⁻ cells could hardly be converted to TEM8^{hi}ALDH⁺ cells (see below Fig. II), indicating that TEM8^{hi}ALDH⁺ marked a less differentiated TIC population. Furthermore, we noticed that TEM8^{hi}ALDH⁻ cells could be converted to the ALDH⁺ cells, while TEM8^{low}ALDH⁺ cells were less converted to TEM8^{hi} cells (see below Fig. II), suggesting cells tend to transform from TEM8^{hi} state to ALDH⁺ state. As you mentioned, we also considered that important regulators of angiogenesis such as nutrition deprivation and hypoxia may be involved in the regulation of the equilibrium. Actually, Chaudhary et al. has found that TEM8 could be selectively upregulated on tumor vasculature in response to growth factor deprivation (Chaudhary et al. Cancer Cell, 2018). Our results also showed that TEM8 was upregulated under hypoxia condition in TNBC cells (see below Fig. III). Even though we have not

discussed in depth in the manuscript, we are going to further explore this in our future work.

Fig. I Immunohistochemistry analyses of TEM8 expression in TEM8^{low} and TEM8^{hi} cell-derived xenografts.

Fig. II The FACS strategy for 4 groups of cells as indicated and purity confirmed by a subsequent flow cytometry detection. After in vitro culture for 7 days, TEM8 and ALDH in the indicated 4 groups of cells were re-examined by flow cytometry.

Fig. III qRT-PCR analysis of the TEM8 mRNA levels in TNBC cells under normoxia or hypoxia.

4) The authors identify a pathway downstream of TEM8 involving RhoC-ROCK1-SMAD5 which intern regulates both stemness and VM. Each of these downstream

effectors has previously been linked to cancer stemness. The authors need to cite these studies. Studies showing the effects of RhoC and ROCK -1 inhibitors on CSC are particularly relevant.

Answer: Thank you very much for your suggestions and sorry about our negligence. We have cited more studies about RhoC (Sang et al., 2016; Rosenthal et al., 2012; Islam et al., 2014), ROCK1 (Wu et al., 2016), and SMAD5 (Genander et al., 2014; Seystahl et al., 2015; Clément et al., 2017) in regulating cancer stemness in the revised manuscript.

5) The authors claim that addition of the ROCK1 inhibitor Y-27632 accentuates the effects of docetaxel chemotherapy. However, Fig5c shows no statically significantly difference in tumor growth between DOC and DOC+ Y-27632.

Answer: Thank you for pointing this out. In our data shown previously, although the tumor growth of DOC + Y-27632 group seemed to be suppressed in comparison to DOC group, no significantly difference was observed due to the individual difference and not enough data collection. Therefore, to reduce the impact of individual difference, we repeated the experiments and have applied six mice for each group in another biological independent experiment, and observed a significantly difference in tumor growth between DOC group and DOC + Y-27632 group (Fig. 5e).

Fig. 5e The effect of Y-27632 treatment (10 mg/kg, intraperitoneally, every three days) combined with docetaxel therapy (10 mg/kg, intraperitoneally, every three days) on TNBC PDX15 tumor growth in NOD/SCID mice (six mice for each group, two sites each mouse, 10^5 cells/site).

6) The authors propose that in ER+ breast cancers, Era regulates the expression of ASB10 to degrade TEM8 in these cells. However, the expression of ER+ is heterogeneous in ER+ cells and in fact in luminal B breast only a few percent of cells express ER. If this was a cell autonomous mechanism, then the cells in these tumors that lacked ER expression should express TEM8 protein. However, this is not consistent with the data shown. To further support their hypothesis, the authors should demonstrate that ASB10 knockdown, results in increased TEM8 protein expression.

Answer: Thanks for your suggestions. Our results suggested that ER α could regulate the expression of ASB10 to degrade TEM8 in luminal cells. Actually, as you mentioned, the expression of ER α is heterogeneous especially in luminal B breast cancer. We did observe a heterogeneous expression of TEM8 in luminal especially luminal B breast cancer patient tumors (see below Fig. IV, relative to Fig. 1b). ER α ⁻ luminal B tumor BC30 expressed relative high level of TEM8 protein when compared with another ER α ⁺ luminal B tumor BC33 (see below Fig. V). Furthermore, we also observed a significant increase of TEM8 protein in ASB10 knockdown luminal type MCF-7 cells (Supplementary Fig. 7n). Thus, we think it is justified to draw the conclusion that ER α trans-activated ASB10 was in part responsible for low expression of TEM8 in ER α ⁺ luminal breast cancer.

Fig. IV H-Score of TEM8 expression in luminal A and luminal B breast cancer patient tissues.

Fig. V Co-stained TEM8 and ER α in ER α ⁻ and ER α ⁺ luminal B breast cancer patient tissues.

Supplementary Fig. 7n Western blotting analysis of TEM8 protein in ASB10-knockdown MCF-7 cells.

7) The manuscript would benefit from further English editing.

Answer: Thanks for your suggestions. For English editing, we have asked a professional organization (the International Science Editing, <http://www.internationalscienceediting.com>) to help, and hopefully it is improved now.

Reviewers' Comments:

Reviewer #1:

Remarks to the Author:

The authors have satisfactorily responded to all the criticisms raised to the previous version

Reviewer #2:

Remarks to the Author:

The authors have satisfactorily addressed the issues raised.

Reviewer #3:

Remarks to the Author:

Xu et al. propose a mechanism where TEM8 triggers activation of RhoC, which then triggers further biological effects via activation of ROCK1. However, there are several aspects in this proposed mechanism that are difficult to reconcile with the current knowledge on Rho GTPases.

Rho GTPases are present in an inactive GDP-bound and an active GTP bound form. Only in their active form they are able to interact with effectors which mediate the biological function. Activation of Rho GTPases is triggered by transient binding to GEFs. All known GEFs for Rho GTPases belong either to the Dbl family (70) or to the DOCK family (11), the latter one however not acting on Rho GTPases of the Rho subfamily as RhoC.

The authors report a direct interaction between RhoC and TEM8. According to current knowledge, this should only be possible if RhoC is in an active GTP bound form and TEM8 a novel RhoC effector. It would further imply that binding of RhoC to TEM8 is changing TEM8 function resulting in a biological effect. Surprisingly, the authors show next that TEM8 overexpression is increasing RhoC activation, indicating that TEM signaling is resulting in RhoC activation. They suggest that the TEM8 binding molecule GNAS is involved in RhoC activation, but I do not understand at all the reasoning for that since GNAS is not a known GEF for Rho GTPases. The authors state that "guanine nucleotide-binding protein G(s) subunit alpha isoforms short (GNAS) showed GTP-binding ability (Fig. 3b)". But why would this be sufficient to be a GEF for RhoC? Later, they write "In summary, ... GNAS activated RhoC through its GTP-binding ability." To me this indicates a complete lack of understanding of GTPases in general.

Moreover, the authors suggest next a direct interaction between GNAS and RhoC. As GEFs by nature only transiently bind to Rho GTPases, which make all pull-downs of course extremely difficult, a direct interaction would rather indicate that GNAS might be another novel RhoC effector than a novel type of RhoC GEF. (Interestingly, dominant negative mutant forms of Rho GTPases prevent Rho GTPase activation by permanently binding to GEFs and thus preventing their activation of wildtype Rho GTPases!). Finally, the authors suggest that RhoC mediates its biological effects in their system by binding to ROCK1 and activating it. It is very interesting how the authors think this will happen if RhoC is at the same time bound to TEM8 and/or GNAS. Rho GTPases are neither known to function while being bound to a GEF (see dnRhoGTPases mentioned above) nor known to bind to two effectors simultaneously.

As a general remark, all WB data should be properly quantitated in addition to showing a representative blot. This is even more important, if a blot is downstream of an IP as in the Rho GTPase activation assay, since the additional experimental procedures increase the assay-to-assay variation.

The authors show that TEM8 overexpression increases RhoC activation (Fig 3I). Then they do KD of GNAS and show that FLAG-TEM8 expression is reduced by about 50%. They do not show how GNAS expression is altered. Still, they interpret the results as a proof that GNAS is mediating TEM8 dependent RhoC activation, although it might also relate to the reduced levels of FLAG-TEM8. Why FLAG-TEM8 is reduced by GNAS KD or whether this might be only caused by assay-to-

assay variation is not discussed. Also, no scrambled shRNA is used as a control for the KD, which is mandatory for knockdown experiments.

We thank you and other editors for the time and efforts on evaluating our submitted manuscript "TEM8 marks neovasculogenic tumor-initiating cells in triple-negative breast cancer". Thank you very much for giving us the opportunity to make the point-to-point responses to your comments. In accordance with the suggestions/comments, appropriate changes have been made in the revised manuscript and highlighted in yellow.

Following are the point-to-point responses to the reviewer's comments.

Reviewer #3 (Remarks to the Author):

Xu et al. propose a mechanism where TEM8 triggers activation of RhoC, which then triggers further biological effects via activation of ROCK1. However, there are several aspects in this proposed mechanism that are difficult to reconcile with the current knowledge on Rho GTPases.

Rho GTPases are present in an inactive GDP-bound and an active GTP bound form. Only in their active form they are able to interact with effectors which mediate the biological function. Activation of Rho GTPases is triggered by transient binding to GEFs. All known GEFs for Rho GTPases belong either to the Dbl family (70) or to the DOCK family (11), the latter one however not acting on Rho GTPases of the Rho subfamily as RhoC.

The authors report a direct interaction between RhoC and TEM8. According to current knowledge, this should only be possible if RhoC is in an active GTP bound form and TEM8 a novel RhoC effector. It would further imply that binding of RhoC to TEM8 is changing TEM8 function resulting in a biological effect. Surprisingly, the authors show next that TEM8 overexpression is increasing RhoC activation, indicating that TEM signaling is resulting in RhoC activation. They suggest that the TEM8 binding molecule GNAS is involved in RhoC activation, but I do not understand at all the reasoning for that since GNAS is not a known GEF for Rho GTPases. The authors state that "guanine nucleotide-binding protein G(s) subunit alpha isoforms short (GNAS) showed GTP-binding ability (Fig. 3b)". But why would this be sufficient to be a GEF for RhoC? Later, they write "In summary, ... GNAS activated RhoC through its GTP-binding ability." To me this indicates a complete lack of understanding of GTPases in general. Moreover, the authors suggest next a direct interaction between GNAS and RhoC. As GEFs by nature only transiently bind to Rho GTPases, which make all pull-downs of course extremely difficult, a direct interaction would rather indicate that GNAS might be another novel RhoC effector than a novel type of RhoC GEF. (Interestingly, dominant negative mutant forms of Rho GTPases prevent Rho GTPase activation by permanently binding to GEFs and thus preventing their activation of wildtype Rho GTPases!).

Answer: Thank you very much for your thoughtful comments. We have repeated the experiments for many times, and we are certain that the expression of TEM8 could lead to the increase of active

RhoC level via its intracellular domain. However, we didn't find any clue indicating that TEM8 would directly activate RhoC. TEM8 protein does not contain the catalytic Dbl-homologous (DH) domain shared by RhoGEFs. As you mentioned above, TEM8 couldn't function as a RhoGEF. Therefore, we considered that TEM8 might increase active RhoC level via other ways. We noticed that TEM8 also interacted with GNAS, which obviously also did not belong to the Dbl family and showed significant influence on the active RhoC level. We found that GNAS knockdown could reverse the increase of active RhoC induced by TEM8 (Fig. 3i), whereas GNAS overexpression could further increase the active RhoC level (Fig. 3j). Besides, we observed the co-localization of TEM8, RhoC and GNAS in TNBC cells (Fig. 3k). The presence of TEM8 protein further enhanced the interaction between RhoC and GNAS (Fig. S6e). Thus, we tended to consider the function of TEM8 like an adaptor protein that promote the interaction between RhoC and GNAS under certain conditions.

Regarding the mechanism of GNAS increasing active RhoC level, we are sorry about the confusing description in previous version of manuscript without further explanation. Although we have found that the stimulative effect of GNAS on active RhoC relied on its GTP binding site (Fig. S6f), GNAS could not function as a RhoGEF since the absence of DH domain. In fact, we have found that GNAS interacted with RhoC via its GTP binding site. Hence, we considered that GNAS might increase the active RhoC level via a non-canonical pathway. It is known that the balance between active form and inactive form of Rho GTPase is generally regulated by RhoGEFs and RhoGAPs. The RhoGEFs induce the release of GDP to be replaced by GTP, while RhoGAPs induce the GTP hydrolysis. The RhoGAP family protein ARHGAP21 is reported to catalyze the conversion of active RhoC to inactive RhoC (Lazarini M. et al, *Biochim Biophys Acta*, 2013). Our further experiments showed that GNAS overexpression could reverse the inactivation of RhoC caused by ARHGAP21 (Fig. 3o). What's more, GNAS could not interact with RhoC protein after deleting the ARHGAP21 binding site of RhoC (Fig. 3p and S6g). Together, these results implied that GNAS functioned neither as a RhoGEF, nor as a RhoGAP, but bound to the RhoC GAP binding site and thus prevented the interaction between RhoC and the RhoGAP, ARHGAP21, therefore leading to the increase of active RhoC level. Similar non-canonical Rho signaling activating pathway could also be seen in Kim O's work which showed that Etk protein increased RhoA activation via binding to RhoA and thus preventing RhoGDI binding to RhoA (Kim O. et al, *J Bio Chem*, 2002). To further elucidate whether GNAS binding to RhoC relied on RhoC activation, we have tried to evaluate the binding ability of GNAS to active RhoC using the constructive active RhoC form RhoC^{G14V}. Our results did not support a stronger binding between GNAS and RhoC^{G14V} (see below Fig. I, shown only for reviewer), which we are going to further explore in our future work.

Fig. S6e The interaction between RhoC and GNAS in the presence or absence of TEM8 protein was analyzed by *in vitro* co-IP assay. FLAG and HA IP bands were quantitated to the corresponding RhoC IP bands.

Fig. 3n Full length GNAS protein (FL) and a GTP binding site (aa357-374)-deleted truncate (Δ GTP) were transfected into 293T cells. The interaction between GNAS and RhoC was analyzed by the Co-IP assay.

Fig. 3o MYC-ARHGAP21 and HA-GNAS were co-transfected into 293T cells as indicated. The activity of RhoC was then determined.

Fig. 3p Full length RhoC protein (FL) and GAP binding site (aa34-39)-deleted truncate (ΔGAP) were co-transfected into 293T cells with HA-GNAS. The interaction between GNAS and RhoC was analyzed by Co-IP assay.

Fig. S6g Full length RhoC (FL) and GAP binding site-deleted truncate (ΔGAP) proteins were purified. The interaction between GNAS and RhoC was analyzed by *in vitro* co-IP assay.

Fig. I RhoC protein (WT) and its constructive active form RhoC^{G14V} (G14V) were co-transfected into 293T cells with HA-GNAS as indicated. The interaction between RhoC and GNAS was analyzed by *in vitro* co-IP assay.

Finally, the authors suggest that RhoC mediates its biological effects in their system by binding to ROCK1 and activating it. It is very interesting how the authors think this will happen if RhoC is at the same time bound to TEM8 and/or GNAS. Rho GTPases are neither known to function while being bound to a GEF (see dnRhoGTPases mentioned above) nor known to bind to two effectors

simultaneously.

Answer: Thanks for your questions. As we demonstrated above, TEM8 and GNAS do not function as RhoGEFs in this study. The function of TEM8 relied on its intracellular domain recruiting GNAS, then GNAS bound to the GAP binding site of RhoC to prevent its GTP hydrolysis by RhoGAP ARHGAP21, which finally increased the RhoC activation. We identified ROCK1 as downstream effector of RhoC. Our results have shown that GNAS knockdown decreased active RhoC, whereas GNAS overexpression increased. But neither GNAS knockdown nor overexpression would block the binding of active RhoC to its effector ROCK1 (Fig. S6b and S6d).

Fig. S6b&S6d GNAS/scrambled shRNA (b) or HA-GNAS (d) were co-transfected into 293T cells with MYC-ROCK1 as indicated. The interaction between ROCK1 and RhoC and the active RhoC level were then determined.

As a general remark, all WB data should be properly quantitated in addition to showing a representative blot. This is even more important, if a blot is downstream of an IP as in the Rho GTPase activation assay, since the additional experimental procedures increase the assay-to-assay variation.

Answer: Thanks for your suggestions. As you suggested, we have quantitated the WB data to its corresponding reference proteins in the revised manuscript. Each GTP-RhoC was quantitated to total RhoC. Each phosphorylated protein was quantitated to its total protein.

The authors show that TEM8 overexpression increases RhoC activation (Fig 3l). Then they do KD of GNAS and show that FLAG-TEM8 expression is reduced by about 50%. They do not show how GNAS expression is altered.

Answer: Thanks for your suggestions. Because there is not a reliable antibody available for us to detect the alteration of endogenous GNAS protein, we have confirmed the knockdown effect of GNAS via the RT-qPCR assay (Fig. S6a and S6c).

Fig. S6a&S6c TEM8 (a) or ROCK1 (c) were co-transfected into 293T cells with GNAS/scrambled shRNA as indicated. mRNA expressions were determined by RT-qPCR.

Still, they interpret the results as a proof that GNAS is mediating TEM8 dependent RhoC activation, although it might also relate to the reduced levels of FLAG-TEM8. Why FLAG-TEM8 is reduced by GNAS KD or whether this might be only caused by assay-to-assay variation is not discussed.

Answer: Thank you for pointing this out. We have repeated this experiment several times, and confirmed the expression of TEM8 by RT-qPCR and WB. Both endogenous TEM8 and FLAG-TEM8 were not reduced by GNAS knockdown when being quantitated with the corresponding reference gene or protein (Fig. 3i and S6a). Reduced level of TEM8 expression by GNAS knockdown was not conclusive. The FLAG-TEM8 knockdown by GNAS shown in the previous version of manuscript is probably due to the assay-to-assay variation.

Fig. 3i GNAS/scrambled shRNA and TEM8 were co-transfected into 293T cells as indicated. The activity of RhoC was then determined.

Fig. S6a GNAS/scrambled shRNA and TEM8 were co-transfected into 293T cells as indicated. mRNA expressions were determined by RT-qPCR.

Also, no scrambled shRNA is used as a control for the KD, which is mandatory for knockdown experiments.

Answer: Thanks for your suggestions. We have used the commercial scrambled shRNA (purchased from Addgene) as a control for all the knockdown experiments.

We hope that we answered all your comments and questions. Thank you very much for your time and support again.

Sincerely,

Suling Liu, PhD, Professor
Shanghai Cancer Center, Fudan University
Shanghai, P.R. China
Phone: 086-21-34771023
Email: suling@fudan.edu.cn

Reviewers' Comments:

Reviewer #3:

Remarks to the Author:

The authors addressed all my comments sufficiently!